# Theta-modulation drives the emergence of connectivity patterns underlying replay in a network model of place cells

Panagiota Theodoni[1,2,3], Bernat Rovira[1,4], Yingxue Wang[5], Alex Roxin[1,6]*

[1]Centre de Recerca Matemàtica, Bellaterra, Spain; [2]New York University Shanghai, Shanghai, China; [3]NYU-ECNU Institute of Brain and Cognitive Science at NYU Shanghai, Shanghai, China; [4]Department of Information and Communication Technologies, Universitat Pompeu Fabra, Barcelona, Spain; [5]Max Planck Florida Institute for Neuroscience, Jupiter, United States; [6]Barcelona Graduate School of Mathematics, Barcelona, Spain

**Abstract** Place cells of the rodent hippocampus fire action potentials when the animal traverses a particular spatial location in any environment. Therefore for any given trajectory one observes a repeatable sequence of place cell activations. When the animal is quiescent or sleeping, one can observe similar sequences of activation known as replay, which underlie the process of memory consolidation. However, it remains unclear how replay is generated. Here we show how a temporally asymmetric plasticity rule during spatial exploration gives rise to spontaneous replay in a model network by shaping the recurrent connectivity to reflect the topology of the learned environment. Crucially, the rate of this encoding is strongly modulated by ongoing rhythms. Oscillations in the theta range optimize learning by generating repeated pre-post pairings on a time-scale commensurate with the window for plasticity, while lower and higher frequencies generate learning rates which are lower by orders of magnitude.
DOI: https://doi.org/10.7554/eLife.37388.001

*For correspondence:
aroxin@crm.cat

**Competing interests:** The authors declare that no competing interests exist.

## Introduction

As an animal explores in any given environment, place cells in the hippocampus fire selectively at particular locations (*O'Keefe and Dostrovsky, 1971*; *O'Keefe and O'Keefe, 1976*; *Harvey et al., 2009*), known as the 'place-fields' of the cells. Furthermore, the place fields of an ensemble of place cells remap to completely new positions with respect to one another if the animal enters a distinct environment (*Muller and Kubie, 1987*; *Kubie and Muller, 1991*; *Bostock et al., 1991*). Sequential place cell activation during exploration therefore acts as a unique fingerprint for each environment, providing information needed for navigation and spatial learning. The spontaneous replay of such sequential activation, which occurs within sharp-wave/ripples (SWRs) during quiet wakefulness (*Foster and Wilson, 2006*; *Karlsson and Frank, 2009*; *Carr et al., 2011*) and sleep (*Wilson and McNaughton, 1994*; *Lee and Wilson, 2002*) suggests that the animal has formed an internal representation of the corresponding environment, presumably during exploration (*Wu and Foster, 2014*). Synaptic plasticity is the clear candidate mechanism for this formation. Nonetheless it remains unclear how changes in the synaptic connectivity are coordinated at the network level in order to generate well-ordered sequences spontaneously. An additional prominent physiological signature of exploratory behavior in the hippocampus is the theta rhythm (4–12 Hz). Decreases in theta power due to lesions of the medial septum strongly reduce performance in spatial-memory based tasks (*Winson, 1978*; *Mitchell et al., 1982*; *Dwyer et al., 2007*) and other hippocampal dependent tasks (*Berry and Thompson, 1979*; *Allen et al., 2002*) although they do not eliminate place fields on

linear tracks (*Brandon et al., 2014*; *Wang et al., 2015*). Despite this, lesioned animals can still reach fixed performance criteria given enough time (*Winson, 1978*; *Berry and Thompson, 1979*). This shows not only that theta is important for learning, but also suggests a potential mechanism. Namely, it may act as a dial for the learning rate, presumably by modulating the network-wide coordination of synaptic plasticity. How this occurs is unknown. Here, we develop a model that explains how synaptic plasticity shapes the patterns of synaptic connectivity in a strongly recurrent hippocampal circuit, such as CA3, as an animal explores a novel environment. We show that a temporally asymmetric spike-timing dependent plasticity (STDP) rule (*Markram et al., 1997*; *Bi and Poo, 1998*) during motion-driven sequential place-cell activity leads to the formation of network structure capable of supporting spontaneous bursts. These bursts occur in the absence of place-field input, that is during awake-quiescence or sleep. Importantly, the spatio-temporal structure of the bursts undergoes a sharp transition during exploration, exhibiting well-ordered replay only after a critical time of exploration in the novel environment. The underlying rate of this plasticity process, and hence the critical time, is strongly modulated through external oscillatory drive: for very low and high frequency the rate is near zero, while for an intermediate range, set by the time-scale of the plasticity rule, it is higher by several orders of magnitude, allowing for learning on realistic time-scales. Our theoretical findings lend support to the hypothesis that the theta rhythm accelerates learning by modulating place-cell activity on a time-scale commensurate with the window for plasticity. This maximizes the growth rate of network-wide patterns of synaptic connectivity which drive spontaneous replay. Finally, we confirm a main prediction from the model using simultaneous recordings of hippocampal place cells from a rat exploring a novel track. Namely, pairwise correlations between cells with neighboring place fields show a pronounced increase over the span of several minutes at the outset of exploration. Furthermore, in a rat in which the medial septum is partially inactivated via muscimol injection, which strongly reduces theta modulation, this increase is not seen.

## Results

### Plasticity during exploration of a novel environment leads to a transition in the structure of sharp-wave events

We modeled the dynamics of hippocampal place cells in a strongly recurrent circuit, such as area CA3, as an animal sequentially explored a series of distinct novel ring-like tracks, *Figure 1a*. The model consisted of recurrently coupled, excitatory stochastic firing rate neurons, each of which received a place-specific external input on any given track, and inhibition was modeled as a global inhibitory feedback, see Materials and methods for model details. To model the global remapping of place fields from one track to another, we randomized the position of the peak input, and as a consequence the place field, of each neuron. In this way, the ordering of cells according to their place field location on one track was random and uncorrelated with their ordering on any other track, see *Figure 1b*.

We were interested in knowing how the exploration of a novel environment affected the pattern of synaptic connectivity between place cells via a temporally asymmetric plasticity rule, and how this in turn shaped the activity. While the plasticity rule we use is formally referred to as 'Spike-timing dependent' (*Kempter et al., 1999*; *Song et al., 2000*; *Pfister and Gerstner, 2006*), our model neurons generate spikes stochastically as Poisson processes. Therefore the exact spike timing itself does not play any role but rather only the time variations in the underlying firing rate. Recent theoretical work has shown that the plasticity induced by a number of plasticity rules, including heuristic STDP rules, is in fact dominated by such firing rate variations when irregular, in-vivo like activity is considered (*Graupner et al., 2016*). We also considered a 'balanced' plasticity rule, for which the degree of potentiation and depression were the same, on average, given constant firing rates. Such a rule is a means of avoiding fully potentiating or fully depressing all synapses on long time-scales, that is it allows for the generation and maintenance of structured synaptic connectivity. Alternatively, we could have considered an unbalanced plasticity rule with additional stabilizing mechanisms (*Zenke et al., 2015*).

In order to study 'typical' place-cell dynamics during exploration we first exposed the network to a series of distinct tracks until the matrix of recurrent synaptic weights no longer depended on its initial state. When the trajectory and velocity of the virtual animal was stochastic and distinct on

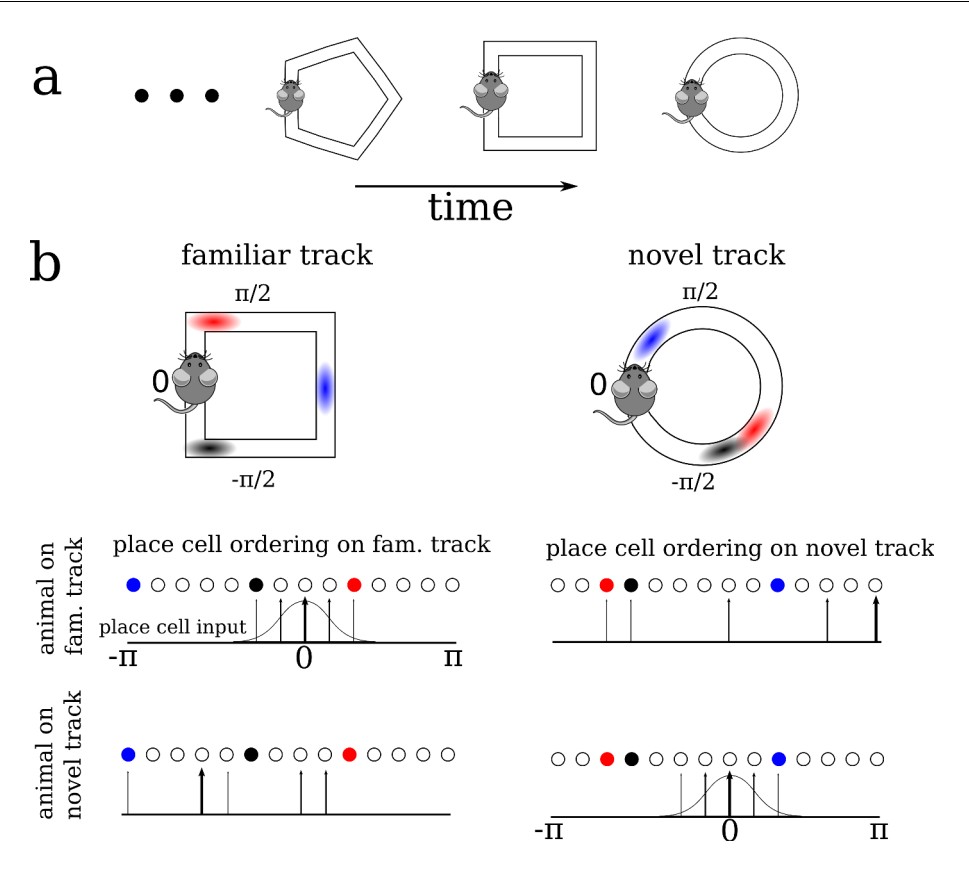

**Figure 1.** A schematic description of the model. (**a**) We model the sequential exploration of a number of distinct linear tracks. (**b**) The network consists of N place cells. The ordering of the place fields is randomly remapped on each track. Therefore, if the cells are properly ordered in any given environment the place field input is represented by a spatially localized bump of activity (upper left and lower right). Sequential activity on a familiar track would look random given an ordering on a novel track, and vice versa (upper right and lower left respectively).

DOI: https://doi.org/10.7554/eLife.37388.002

different tracks, the resulting evolution of the synaptic weights also exhibited some variability from track to track, see *Figure 2—figure supplement 1*. In simulations with constant velocity this variability largely vanished, see *Figure 2—figure supplement 2a*. After exploring 10 tracks, we exposed the network to another novel track and studied the dynamics in detail; because the exploration process had already become largely stereotyped (despite some variability), the dynamics on the novel track reflected what would be seen qualitatively during the exploration of any other novel track in the future, that is it was dependent only on the learning process itself and not the initial state of the synaptic weight matrix.

In our simulations, we modeled the movement of a virtual animal around the track by varying the external inputs to place cells. Specifically, the external input was maximal for cells with place fields at the animal's current position and minimal for cells with place fields at the opposite side of the track, with the input decaying smoothly for intermediate locations like a cosine curve, see Materials and methods for details. We modeled the motion of the animal by taking the velocity to be an Ornstein-Uhlenbeck process with a time constant of 10 s; this lead to realistic trajectories with changes of direction, see *Figure 2—figure supplement 3*. (We will use the metaphor of a virtual animal for conceptual ease, although we really mean that we moved the bump-like external input to our model neurons). Every three minutes of simulation time we stopped the animal at its current location for three seconds and removed the place field input. This led to spontaneous bursting via a synaptic depression-dependent mechanism (*Romani and Tsodyks, 2015*), reminiscent of sharp-waves seen

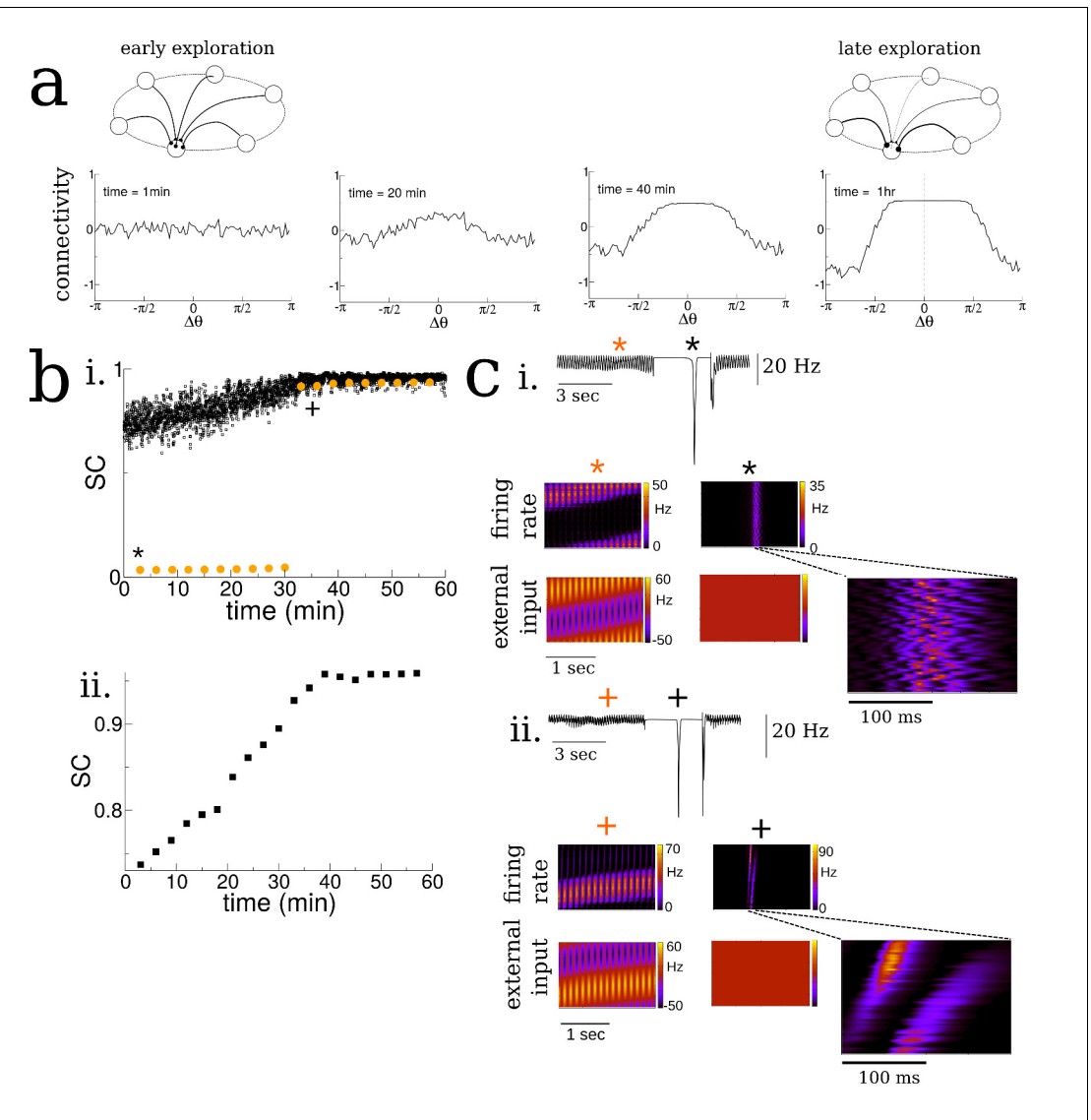

**Figure 2.** Spatial exploration gives rise to plasticity-dependent emergence of spatio-temporal structure in spontaneous activity. (a) Snapshots of the connectivity during the exploration of a novel environment. During early exploration, the connectivity is not correlated with the place-field ordering, while during late exploration cells with neighboring place-fields are more strongly connected. (b) i. The mean cross-correlation of the activity of place cells with adjacent place fields on a novel track during exploration (sequential correlation SC). Black: SC during active exploration (theta-activity), Orange: SC during spontaneous bursts. Note the sharp transition in the SC of burst activity. ii. The SC of the total activity binned into 3 min intervals shows a steady increase preceding the transition. (c) Burst activity exhibits replay after a critical period. i. Early exploration. Top: average input to place cells before (blue star), during (black star) and after period of 'quiet wakefulness'. Bottom: space-time plots of the place cell input and firing rate. Note the disordered spatio-temporal structure of the burst activity. ii. Later exploration. After a critical transition time bursts exhibit sequential replay of activity from the novel track. Note the sequential structure of the burst activity.

DOI: https://doi.org/10.7554/eLife.37388.003

The following figure supplements are available for figure 2:

**Figure supplement 1.** The evolution of the recurrent connectivity during exploration of 10 distinct tracks.
DOI: https://doi.org/10.7554/eLife.37388.004

**Figure supplement 2.** The emergence of replay for unidirectional motion.
DOI: https://doi.org/10.7554/eLife.37388.005

*Figure 2 continued on next page*

*Figure 2 continued*

**Figure supplement 3.** The growth of the odd mode of the recurrent connectivity is determined by the bias in the motion of the virtual animal.
DOI: https://doi.org/10.7554/eLife.37388.006
**Figure supplement 4.** Heterogeneity in place-cell activity does not qualitatively alter the transition to replay.
DOI: https://doi.org/10.7554/eLife.37388.007
**Figure supplement 5.** Theta sequences and phase precession emerge over time.
DOI: https://doi.org/10.7554/eLife.37388.008
**Figure supplement 6.** Forward replay occurs spontaneously, but backward replay requires location-specific input.
DOI: https://doi.org/10.7554/eLife.37388.010
**Figure supplement 7.** The degree of 'burstiness' of spontaneous activity can be modulated by a global external input.
DOI: https://doi.org/10.7554/eLife.37388.011
**Figure supplement 8.** Changes in the recurrent connectivity stabilize in an extended, 10 hr simulation.
DOI: https://doi.org/10.7554/eLife.37388.009

during awake quiescence and sleep, see *Figure 2*. Neither the bursting, nor the learning process itself depended qualitatively on the exact amount of time the virtual animal spent moving versus staying still. Synaptic plasticity was allowed to occur during both theta activity and sharp-waves, that is it was never 'turned-off'.

As the virtual animal explored the track, the sequential activation of the place cells led to changes in the recurrent connectivity via the plasticity rule, *Figure 2a*. Whereas the connectivity between cells was initially unstructured on the novel track, it evolved over the span of minutes to tens of minutes such that cells with nearby place fields were more strongly connected than ones with disparate place fields. Furthermore the resulting connectivity was slightly asymmetric, reflecting the bias of the animal's motion, see *Figure 2—figure supplement 3*. Although the changes we observed in the recurrent connectivity, see *Figure 2a*, and concomitant changes in place cell activity were continuous and smooth during the duration of the simulation, there was a dramatic and sharp transition in the structure of the burst activity during awake quiescence. We quantified this transition by measuring the mean pairwise cross-correlation of cells with neighboring place fields on the novel track. Such cells are 'neighbors' only on the novel track and not on any other track by virtue of the global remapping or randomization of place-fields. We call this measure the sequential correlation (SC) because it is high when there is a properly ordered sequence of place-cell activations on any given track. The SC during theta-activity (exploration) was already non-zero at the outset of the simulation by virtue of the external place-field input which generates sequential activity, black squares *Figure 2b*. On the other hand, the SC was initially near zero during spontaneous bursts, red circles *Figure 2b*. Interestingly the SC abruptly increased during burst activity at a critical time and remained elevated for the duration of the simulation. This transition occurred even in simulations where we included strong heterogeneity in place field tuning, see *Figure 2—figure supplement 4*. Note that the SC for theta-activity also showed a steady increase leading up to this transition, as did the SC when the total activity was taken into account, without discerning between epochs of movement or awake quiescence, see *Figure 2b* inset. This steady increase is due to the emergence of so-called theta sequences (*Foster and Wilson, 2007*). These are bursts of place-cell activity which sweep ahead of the animal's position on every theta cycle during awake exploration, see *Figure 2—figure supplement 5*. The emergence of theta sequences here also coincides with a precession of the phase of the underlying firing rate with the ongoing theta rhythm, see *Figure 2—figure supplement 5f* and discussion for more detail.

Space-time plots of place-cell activity show that the abrupt increase in SC coincides with a transition in the spatio-temporal structure of the bursts: it is initially uncorrelated with the ordering of cells on the novel track, *Figure 2c (i)*, and after the transition there is a clear sequential activation reminiscent of so-called 'replay' activity, *Figure 2c (ii)*. In fact, before the transition the bursts are highly structured, but this structure is correlated with a previously explored track and not the novel one. If the learning process was carried on for much longer times all relevant network properties, such as connectivity and mean firing rates, saturated, see *Figure 2—figure supplement 8*.

## The transition in the structure of bursts is strongly dependent on the shape of the plasticity rule and the frequency of 'theta' modulation

We hypothesized that changes in the recurrent connections between place cells in our model during exploration were shaping the spontaneous activity and driving the transition to replay-like activity. The connectivity profile at any point in time could be decomposed into a series of spatial Fourier modes, *Figure 3a*. We tracked these modes in time and discovered that the transition in burst activity always occurred when the coefficient of the first even mode reached a critical value. Specifically, we conducted additional simulations in which we increased the firing rate of the feedforward inputs which drove place cell activity which in turn led the transition in burst activity to occur at earlier times, see shaded regions in *Figure 3b*. In fact, a theoretical analysis of our model shows that there is an instability of the spontaneous activity to bursts when the even mode of the connectivity times the gain in the neuronal transfer function is greater than a critical value, see Materials and methods. In our simulations the gain in the transfer function was nearly always the same during bursts because it depended on the mean input to the network during quiescence, which was constant. Therefore it was principally the even mode of the recurrent connectivity which determined the time of the transition. For this reason, the transition to replay occurred when the coefficient of the even mode reached a critical value as seen in *Figure 3c*. Note that there was no such dependence on the odd mode (dotted lines in *Figure 3c*).

We then asked how the growth of the even mode depended on the details of the plasticity rule and the frequency of periodic modulation of the place cell activity. We found that the growth of the even mode depended strongly on the frequency, peaking at an intermediate range and decreasing by orders of magnitude at low and high frequencies, *Figure 4ai*. This occurred independent of the details of the animal's motion, for example compare *Figure 4ai* and *Figure 2—figure supplement 2e* for irregular versus purely clockwise motion respectively. On the other hand the odd mode, which

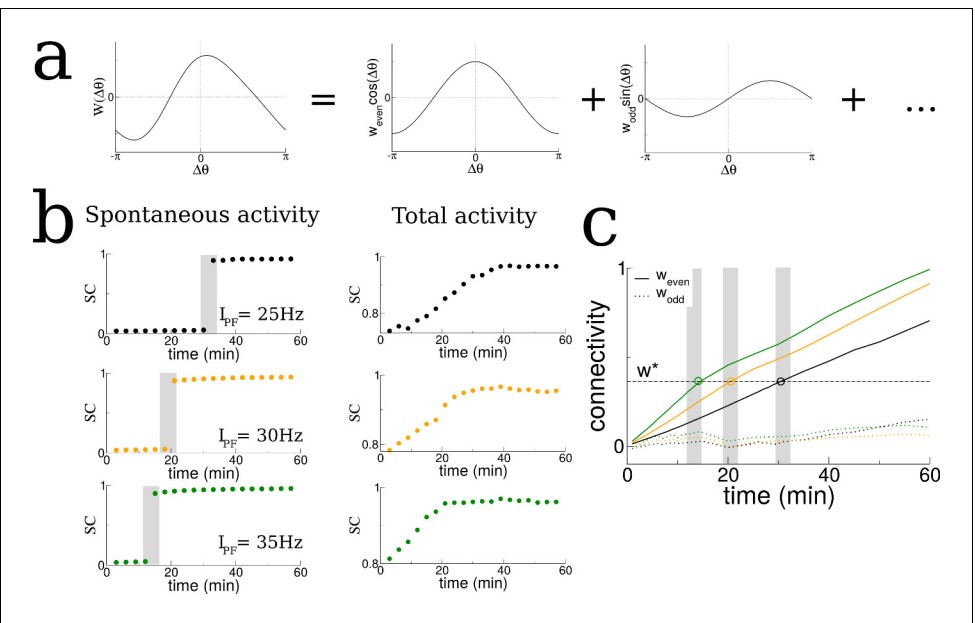

**Figure 3.** The transition in the structure of burst activity occurs when the even component of the recurrent connectivity reaches a critical value. This mode grows only for STDP rules with dominant potentiation, and grows fastest when periodic modulation is in the theta range. (a) The profile of the recurrent connectivity at any point in time can be decomposed into a series of even (cosine) and odd (sine) modes. Only the first two modes are shown. (b) As the amplitude of the place-field input is increased from 25 Hz (black), 30 Hz (orange) and 35 Hz (green) the transition shifts to earlier times. The SC of the SWR activity shows a sharp transition (shaded grey bars), while the total activity displays a smooth increase leading up to the transition. (c) In all cases the value of the even mode reaches approximately the same critical value at the time of the transition (shaded grey bars). The growth of the odd mode (dotted lines) is much more irregular, and its value does not correlate with the transition time.
DOI: https://doi.org/10.7554/eLife.37388.012

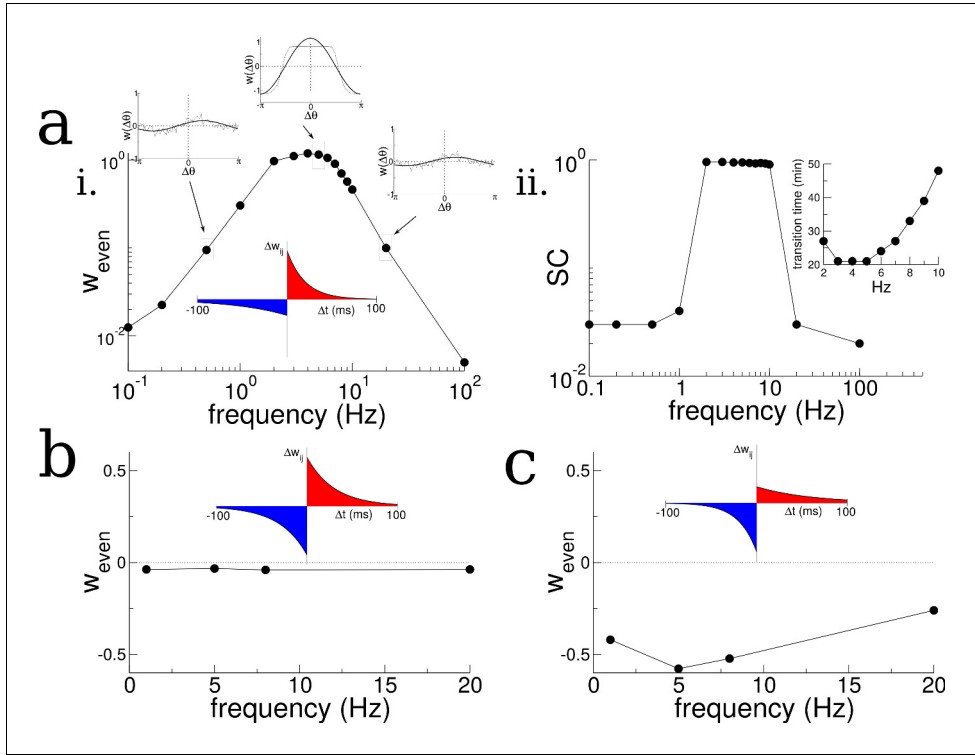

**Figure 4.** Theta-modulation accelerates emergence of connectivity mode which drives replay. (a) A temporal asymmetric plasticity rule with dominant potentiation at short latencies. i. The amplitude of the even mode of the connectivity after 1 hr of exploration, as a function of the modulation frequency. The even mode grows maximally over a range of approximately 1–10 Hz and it is strongly attenuated at lower and higher frequencies. Note the logarithmic scale. On the other hand, the growth of the odd mode is largely independent of the modulation frequency, see text for details. ii. The degree of SC of spontaneous bursting after 1 hr of exploration. Replay occurs only when the frequency lies between 2–10 Hz. Inset: The time at which a transition in the SC takes place as a function of frequency, for times up to 1 hr. (b) A purely anti-symmetric plasticity rule leads to recurrent connectivity which has only odd Fourier modes. There is no increase or transition in the SC of bursts in these simulations even after 1 hr. (c) An asymmetric plasticity rule with depression dominating at short latencies leads to connectivity with a negative amplitude of the even mode, that is recurrent excitation is weaker between pairs of place cells with overlapping place fields than those with widely separated place fields. In this case bursts are actually completely suppressed (not shown).

DOI: https://doi.org/10.7554/eLife.37388.013

represents a directional bias in the recurrent connectivity, closely tracks the cumulative bias in the animal's motion, see *Figure 2—figure supplement 3*, and is independent of the forcing frequency to leading order, see Materials and methods for details. The strong dependence of the growth of the even mode on frequency, meant that, for simulations of up to 1 hr, transitions in the burst activity were observed only when the frequency was in the range of 2–10 Hz, *Figure 4aii*. Furthermore, the even mode only grew, and transitions only occurred, when the plasticity rule had dominant potentiation at short latencies. For a perfectly anti-symmetric plasticity rule, and for a rule with dominant inhibition at short latencies, the even mode did not change and even decreased, respectively, *Figure 4b and c*. In the latter case bursts were suppressed entirely (not shown).

## A self-consistent theory to explain how the interplay between theta-modulation and the plasticity rule govern changes in recurrent connectivity

We sought to understand the mechanism underlying the evolution of the recurrent connectivity seen in simulations by studying a continuum version of a network of place cells, which can be written

$$\tau\dot{r} = -r + \phi\left(\frac{1}{2\pi}\int_{-\pi}^{\pi} d\theta' w(\theta,\theta')r(\theta',t) + I(\theta,t)\right), \tag{1}$$

where $r(\theta,t)$ is the firing rate of a place cell with place field centered at a location $\theta$, $w(\theta,\theta')$ is the synaptic weight from a cell at a position $\theta'$ to a cell at a position $\theta$, and $I(\theta,t)$ is the external input which has the form

$$I(\theta,t) = I_0 + I_{\mathrm{PF}}\cos(\theta - x(t))(1 + I_{\mathrm{theta}}\cos(2\pi ft)). \tag{2}$$

The position of the virtual animal is given by $x(t)$, and the corresponding place-field input is modulated with a frequency $f$. This type of multiplicative theta modulation is seen in intracellular recordings in-vivo, for example see *Figures 1* and *5* in (*Harvey et al., 2009*) and *Figure 4* in (*Lee et al., 2012*).

To model the evolution of the recurrent connectivity we made use of the fact that the step-wise increases in synaptic strength due to the plasticity rule can be approximated as a smooth process as long as plasticity occurs much more slowly than the firing rate dynamics. When this is the case, the rate of change of the synaptic weight from place cell with place field centered at $\theta'$ to one with place field at $\theta$ can be written as

$$\dot{w}(\theta,\theta') = \int_{-\infty}^{\infty} dTK(T)r(\theta,t)r(\theta',t+T), \tag{3}$$

where $K(T)$ is the change in the synaptic weight according to the plasticity rule given a spike pair with latency $T$ (*Kempter et al., 1999*) and see Materials and methods. This equation reflects the fact that the total change in the synaptic weight is the sum of all the pairwise contributions from the pre- and post-synaptic cells, with each pair of spikes weighted by the plasticity rule with the appropriate latency. (*Equations 1–3*) represent a self-consistent model for the co-evolution of the firing rates and synaptic weights in the network.

In order to derive an analytical solution we first assume that the neuronal transfer function $\phi$ is linear. We then make the assumption of slowly evolving synaptic weights explicit by scaling the amplitudes of the potentiations and depressions from the plasticity rule by a small parameter. The upshot is that the connectivity evolves to leading order only on a slow time scale, much slower than the fast neuronal dynamics. Furthermore, we know from numerical simulations that after sufficient exploration the probability of connection between any two cells depends on average only on the difference in place field locations. Therefore, by averaging the connectivity over the fast time $t$ we can write

$$\langle\dot{w}(\theta,\theta')\rangle_t = \langle\dot{w}(\theta-\theta')\rangle_t = \langle\dot{w}_{even}\rangle_t\cos(\theta-\theta') + \langle\dot{w}_{odd}\rangle_t\sin(\theta-\theta') \tag{4}$$

where the growth rates of the even and odd modes $\langle\dot{w}_{even}\rangle_t$ and $\langle\dot{w}_{odd}\rangle_t$ are functions of the plasticity rule parameters, the velocity of the animal and the frequency of periodic modulation, see Materials and methods for details. It turns out it is possible to understand these dependencies intuitively and comprehensively without having to study the analytical formulas. Specifically, if we wish to isolate the growth rate of the even mode, which is responsible for driving the emergence of replay in the burst, we can consider place cell pairs where $\theta = \theta'$, that is pairs with overlapping place fields. When this is the case we can combine (*Equations 3 and 4*) to yield

$$\langle\dot{w}_{even}\rangle_t = \int_{-\infty}^{\infty} dTK(T)AC(T), \tag{5}$$

where $AC(T) = \langle r(\theta,t)r(\theta,t+T)\rangle_t$ is the autocorrelation (AC) of the place-cell activity. Note that despite the similarity in form between (*Equation 5*) and (*Equation 3*), the biological interpretation of the two is quite distinct. (*Equation 3*) describes the changes in the strength of a specific synapse, that from a cell with place-field centered at a position $\theta'$ onto a cell with place-field centered at a position $\theta$. The evolution of this single synapse depends only on the sum of the changes from all spike pairs from these two cells. If we averaged this equation over fast fluctuations in time we would find that the slow evolution of the synapse depends on the cross-correlation of the activity of the

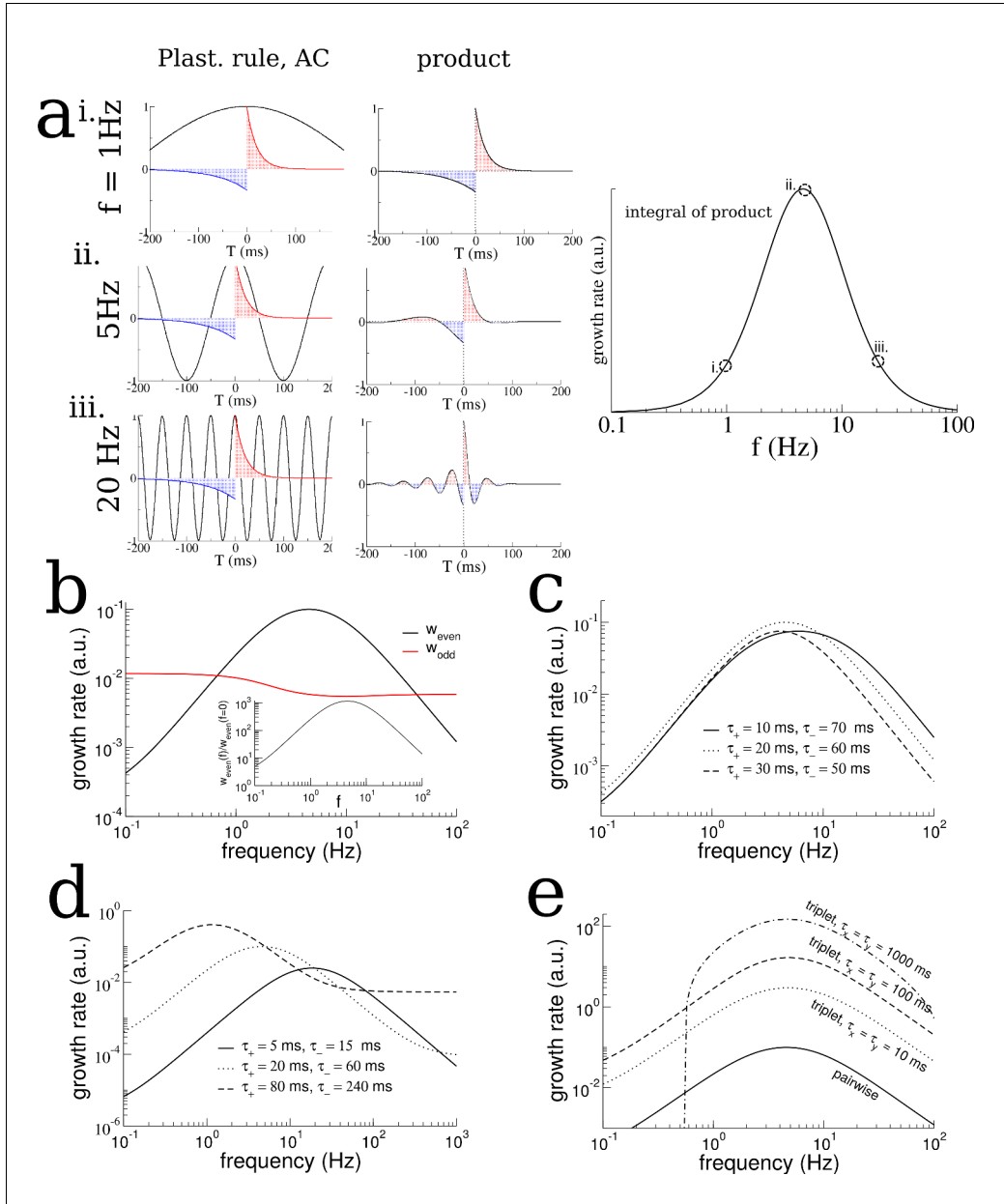

**Figure 5.** Analysis of a linear firing rate model with plasticity reveals the mechanism by which the plasticity rule and periodic modulation shape connectivity. (a) The growth rate of the symmetric mode is proportional to the integral of the product of the autocorrelation (AC) of place cell activity with the plasticity window (PW). Left: The AC of place cell activity overlaid on the PW with dominant potentiation at short latencies. Middle: Product of the PW and the AC. Right: The growth rate of the symmetric (cosine) mode of the connectivity as a function of frequency. i. When the activity is modulated at 1 Hz (top), the product returns nearly the original plasticity window, which being balanced yields near-zero growth rate. ii. For 5 Hz the potentiating lobe is maintained and some of the depression lobe changes sign and becomes potentiating leading to higher growth rate. iii. For 20 Hz the plasticity window undergoes sign reversals at a rate faster than the width of the lobes, meaning the integral is again near zero. (b) The growth rate of the even (cosine) and odd (sine) spatial Fourier coefficients as a function of the modulation frequency (black curve is the same as in (a) except on a log-log scale). Inset: The growth rate of the even mode normalized by its value for no periodic modulation. Rule parameters are $A_+ = 0.1$, $\tau_+ = 20$ ms, $A_- = 0.1/3$, $\tau_- = 60$ ms (c) Increased growth rate in the theta range does not require fine tuning. (d) The frequency at which growth is maximal depends on the overall width of the plasticity window. Broader windows favor slower frequencies. (e) A triplet rule increases the growth rate at all frequencies compared to the pairwise rule, but does not significantly shift the optimal frequency range. The parameters for pairwise interactions are as before. The

*Figure 5 continued on next page*

*Figure 5 continued*

time constants for triplet interactions are indicated on the figure, while the remaining parameters are chosen to make the rule balanced.

DOI: https://doi.org/10.7554/eLife.37388.014

two neurons times the STDP window. There is an equation like this for every cell pair in the network; clearly for a large network it is not feasible to solve these equations self-consistently. On the other hand, (*Equation 5*) does not describe the changes in strength of a single synapse. Rather, it describes changes in the strength of a particular *pattern* of synaptic connectivity in the network. This pattern is one in which cells with highly overlapping place fields have strong and symmetric recurrent connectivity. Furthermore the strength of the synaptic connections decays smoothly with the difference between place field locations. In our theoretical model, in the limit of a large network, the cross-correlation of very nearby cells is simply given by the auto-correlation, which is why it appears in (*Equation 5*). It is perhaps remarkable that using purely local information, namely the AC of place cell activity, one can infer the growth rate of a network-wide mode of synaptic connectivity. The key assumption that makes this possible is that the synaptic weight between any two cells should depend only on the difference in place-field location and not on the absolute position, (*Equation 4*). Therefore the growth rate of the even mode is found by multiplying the AC of place-cell activity by the window for plasticity, and integrating. Because the effect of periodic modulation on the AC is straightforward, we can determine graphically how the frequency of modulation interacts with the plasticity rule to drive changes in the burst structure.

We first note that if there is no periodic modulation of place-cell activity then the AC will simply reflect the movement of the animal. This will lead to a very broad AC compared to the time-scale of plasticity. For example, if we assume that the width of the place field is a fraction of the track length (as in our model), then a rat running between 5 and 50 cm/sec on a multi-meter track would have an AC which decays on the order of between seconds and tens of seconds. Therefore, the AC is essentially constant compared to a typical plasticity window of between tens and hundreds of milliseconds, and the integral in (*Equation 5*) will give nearly zero. Periodically modulating place-cell activity will increase the growth rate as long as potentiation is dominant at short latencies, *Figure 5a*. Slow modulation will bias the integral in (*Equation 5*) toward the potentiation lobe of the STDP (*Figure 5a* left and middle, top), while in an optimal range of frequencies the peaks and troughs of the AC will maximally capture potentiation and flip the sign of the depression lobe (*Figure 5a* left and middle, middle). Finally, at higher frequencies the plasticity rule undergoes multiple sign flips on a fast time scale, which again gives a near zero value for (*Equation 5*) (*Figure 5a*, left and middle, bottom). This means that the maximal growth rate of the even mode occurs for an intermediate range of frequencies: those which modulate place-cell activity on a time scale commensurate with the window for plasticity, *Figure 5a* right. Note that this has nothing to do with the overall rate of plasticity, which is the same independent of the modulation frequency. That is, even if the AC is flat, recurrent synapses undergo large numbers of potentiations and depressions. Rather, periodic modulation serves to organize the structure of synaptic plasticity at the network level by preferentially strengthening connections between place-cells with overlapping or nearby place field and weakening others. The optimal range of frequencies for growth of the even mode depends only weakly on the ratio of the width of the potentiation to that of the depression lobe, *Figure 5c*, but significantly on the total width, *Figure 5d*. Allowing for triplet interactions (as opposed to just pairwise) in the plasticity rule increases the overall growth rate but does not alter the range of optimal frequencies, *Figure 5e*. On the other hand, the theory predicts that the growth rate of the odd mode is only weakly dependent on the modulation frequency (*Figure 5b*) as is seen in simulations (*Figure 2—figure supplement 2e*), and can be understood by considering (*Equation 3*) with $\theta - \theta' = \pi/2$. In this case the growth rate depends on the product of the plasticity rule with the cross-correlation (CC) of cells with disparate place fields. When there is an overall direction bias in motion then the CC will have peak shifted from zero-lag and the product with the STDP rule reliably gives a positive (or negative) growth rate. When there is very little motion bias the CC will be nearly flat, yielding little growth in the odd mode and the resulting connectivity will be highly symmetric.

It is also clear from (*Equation 5*) that a perfectly anti-symmetric plasticity rule will lead to no growth in the even mode as the product of the rule and the AC will itself be an odd function, see *Figure 4b*. A rule with dominant depression at short latencies will similarly yield a growth rate which is precisely the inverse of that shown in *Figure 5a*. This causes a decay in the even mode resulting in a connectivity pattern with locally dominant inhibition, see *Figure 4c*.

Finally, to go beyond this qualitative treatment and find the exact dependence of the evolution of the synaptic weights on model parameters requires solving (*Equations 1–3*) self-consistently, see (*Equations 43–45*) in Materials and methods for the case when $v$ is constant. These are the equations we use to generate the curves in *Figure 5*. When the AC of the place cell activity is predominantly shaped by the modulation frequency $f$, compared to the motion of the animal (strictly speaking when $v/(2\pi f) \ll 1$, which is the case already for $f > 1$ Hz and for reasonable values of $v$) then the connectivity evolves to leading order as

$$\langle \dot{w}_{even} \rangle_t \propto I_{PF}^2 I_{theta}^2 \left[ \frac{1}{1 + 4\pi^2 \tau_+^2 f^2} - \frac{1}{1 + 4\pi^2 \tau_-^2 f^2} \right], \tag{6}$$

$$\langle \dot{w}_{odd} \rangle_t \propto I_{PF}^2 v \left[ \frac{\tau_+}{1 + \tau_+^2 v^2} + \frac{\tau_-}{1 + \tau_-^2 v^2} \right]. \tag{7}$$

*Equation 6* shows clearly that the growth of the even mode, which drives the emergence of replay, is proportional to strength of oscillatory modulation $I_{theta}$. Additionally it can be seen on inspection that the modulation frequency itself plays a crucial role in setting the growth rate, and peaks at intermediate values. On the other hand, the growth of the odd mode, which represents the degree of asymmetry in the recurrent connectivity, is driven entirely by the motion of the animal to leading order. In the case of constant motion, studied here, it depends only on the velocity of the animal.

## Sparse coding allows for the replay of activity from multiple explored tracks during bursts

The spatio-temporal structure of sharp-wave-like bursts in our model reflected the sequential ordering of place fields on the most recently explored track; correlation with earlier tracks was erased or greatly reduced. In reality, only a fraction of place cells have well-defined place fields in any given environment, that is coding is sparse (*Bostock et al., 1991*; *Fyhn et al., 2007*). We incorporated this sparse-coding strategy to our model by providing place-field input to only one half of the total neurons in the network; the other half received constant input. These place cells were chosen randomly from one track to the next and place field locations were assigned randomly as before. Therefore the overlap in the population of place-cells between any two tracks was also fifty percent. We then allowed the network to evolve by having the virtual animal explore thirty distinct tracks, each for 1 hr of simulation time, as before. The resulting matrix of synaptic connections was correlated with the ordering of place-cells in several past explored environments, *Figure 6a*. The amplitude of the even mode, which is responsible for generating spontaneous replay, decayed roughly exponentially as a function of the recency of the explored track, *Figure 6a* inset. This suggested that the replay dynamics in the network with sparse coding might be correlated with several previously explored tracks simultaneously (*Romani and Tsodyks, 2015*). Indeed, when the network was driven with a global, non-selective input, the replay was strongly correlated with the past two explored environments, *Figure 6b*, whereas the correlation with earlier environments was negligible. On the other hand, when the constant external drive was selective to the subset of place-cells on any given track, replay activity was robustly correlated with activity on that track only, *Figure 6c*. Such selective input may originate in cortical circuits which store environment-specific sensory information as stable patterns of activity; these patterns correspond to attracting fixed points in network models of long-term memory storage and memory recall (*Hopfield, 1982*; *Tsodyks and Feigel'man, 1988*; *Recanatesi et al., 2015*). Spontaneous switching between cortical representations during slow-wave activity would therefore engage distinct hippocampal replay patterns, which in turn could strengthen and consolidate the corresponding cortical memory trace.

Therefore, sparse coding causes the synaptic connectivity matrix to be simultaneously correlated with place field orderings from multiple tracks and allows for robust replay of activity from those

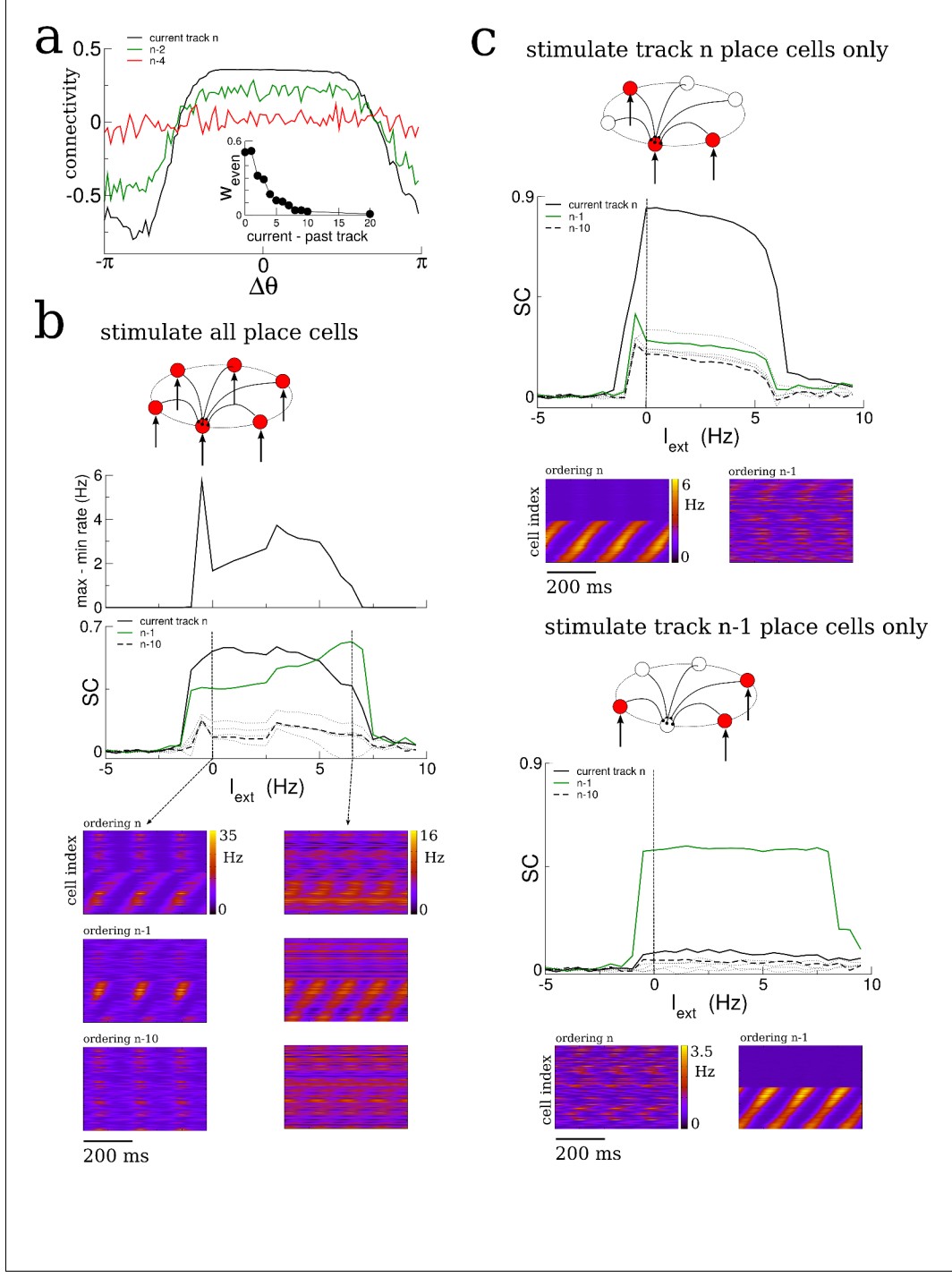

**Figure 6.** Sparse coding allows for the replay of multiple previously experienced environments. (**a**) The connectivity profile after exploration of 30 environments in a network in which one half of the neurons are place cells on any given track. The same connectivity can be visualized using the ordering of the place cells on the most recently explored track (track n, black curve) or using the ordering of past explored tracks (green and red curves). The spatial structure which emerges during exploration on any given track eventually gets overwritten as new tracks are explored. Nonetheless the connectivity stores structure from several past explored tracks simultaneously. Inset: The amplitude of the even Fourier mode as a function of the recency of the corresponding track. (**b**) A global stimulus applied to all the neurons in the network generates replay which is highly correlated with the past two tracks. (**c**) Selective stimulation of only those neurons which were place cells in the most recently

*Figure 6 continued on next page*

*Figure 6 continued*

explored track (top) or next-to-last track (bottom) generates replay which is highly correlated only with the corresponding environment. N = 200 neurons in all simulations.

DOI: https://doi.org/10.7554/eLife.37388.015

tracks given appropriate inputs. The number of different environments which could be simultaneously encoded in replay depended on model details, particularly the coding level, that is the fraction of active place cells in any given environment (not shown) (*Battaglia and Treves, 1998*).

## Experimental evidence for a transition to replay

Pairwise reactivations of CA1 place-cells with overlapping place fields during awake SWRs improve during exploration; they are stronger during late exploration than early exploration (*O'Neill et al., 2006*). This holds true not only for pairwise correlations, but also reactivations of entire neuronal ensembles, at least on linear tracks (*Jackson et al., 2006*). More recent work has shown that significant replay events during awake SWRs in rats running along a three-arm maze emerge abruptly only after a certain number of runs (*Wu and Foster, 2014*). These results are consistent with our model predictions. We additionally sought to directly test for a time-resolved increase in SC in neuronal data. We looked for this increase in multi-unit recordings of place-cell activity from the hippocampus of rats exploring novel, periodic tracks (*Wang et al., 2015*).

We first identified cells with well-defined place fields by extracting the coefficients and phase of the first two spatial Fourier modes of their time-averaged activity as a function of the normalized distance along the track (in radians), see *Figure 7—figure supplements 1–3*. We kept only those cells for which the ratio of the coefficients, the Tuning Index (TI), exceeded the threshold of one, indicating strong spatially selective activity, see methods for details. We then ordered the cells according to their phase (approximately the position of peak firing). The SC of activity over the total duration of the experiment using this ordering was significantly higher than 5000 randomly reshuffled orderings which on average gave SC = 0, see *Figure 7a and b*.

When the medial septum (MS) was inactivated via muscimol the SC did not exhibit any dynamics as a function of time, *Figure 7c*. However, once the animal recovered from the muscimol the SC using the proper phase ordering exhibited an initial ramp over the first ten minutes of exploration and then remained high (significant difference between first point and all others, t-test with multiple-comparison Bonferroni correction, p-values<0.004 and between second and third points), see *Figure 7d*, solid circles. This is consistent with the model result which showed a similar ramping increase when the total activity (and not just bursts) was considered, see for example *Figure 2bii*. On the other hand, the SC computed for shuffled phases showed no dynamics and remained close to zero, see *Figure 7c,d*, open squares. This indicates that there are no global changes in neuronal correlations during exploration, which could occur, for example, due to slow changes in theta modulation or neuronal excitability. Rather, there is a sustained increase in pairwise correlations only between those neurons which encode nearby places, and only when strong theta modulation is present. This finding also held when we considered the more lax criterion $TI \geq 1/2$, see *Figure 7—figure supplement 4* (rat A991). In a second animal there was an insufficient number of well-tuned units to repeat the analysis, see *Figure 7—figure supplement 4* (rat A992). One possible confound in attributing this increase in correlation to theta modulation alone, is the fact that the firing rates of place-cells during MS inactivation were lower on average than during the post-muscimol experiment. Lower firing rates in the computational model lead to lower rates of plasticity. However, this difference in firing rates was not significant for the well-tuned neurons shown in *Figure 7* ($TI \geq 1$), (t-test, p=0.30). An initial, sustained increase in SC was also observed in data from a separate experiment in which an animal was first exposed to a novel hexagonal track, *Figure 7e* (difference between first point and all others, t-test with correction for multiple comparison, p-values<10⁻⁶). Furthermore, no such increase was found on subsequent sessions on the same track, indicating that the change in correlation only occurred when the environment was novel. This result held when we considered the more lax criterion $TI \geq 1/2$ and also when all units were used, regardless of tuning, see *Figure 7—figure supplement 3a* (rat A992). In a second rat there were too few well-tuned cells to use the criterion $TI \geq 1$, but an initial and sustained increase in SC was also seen using all units, although this

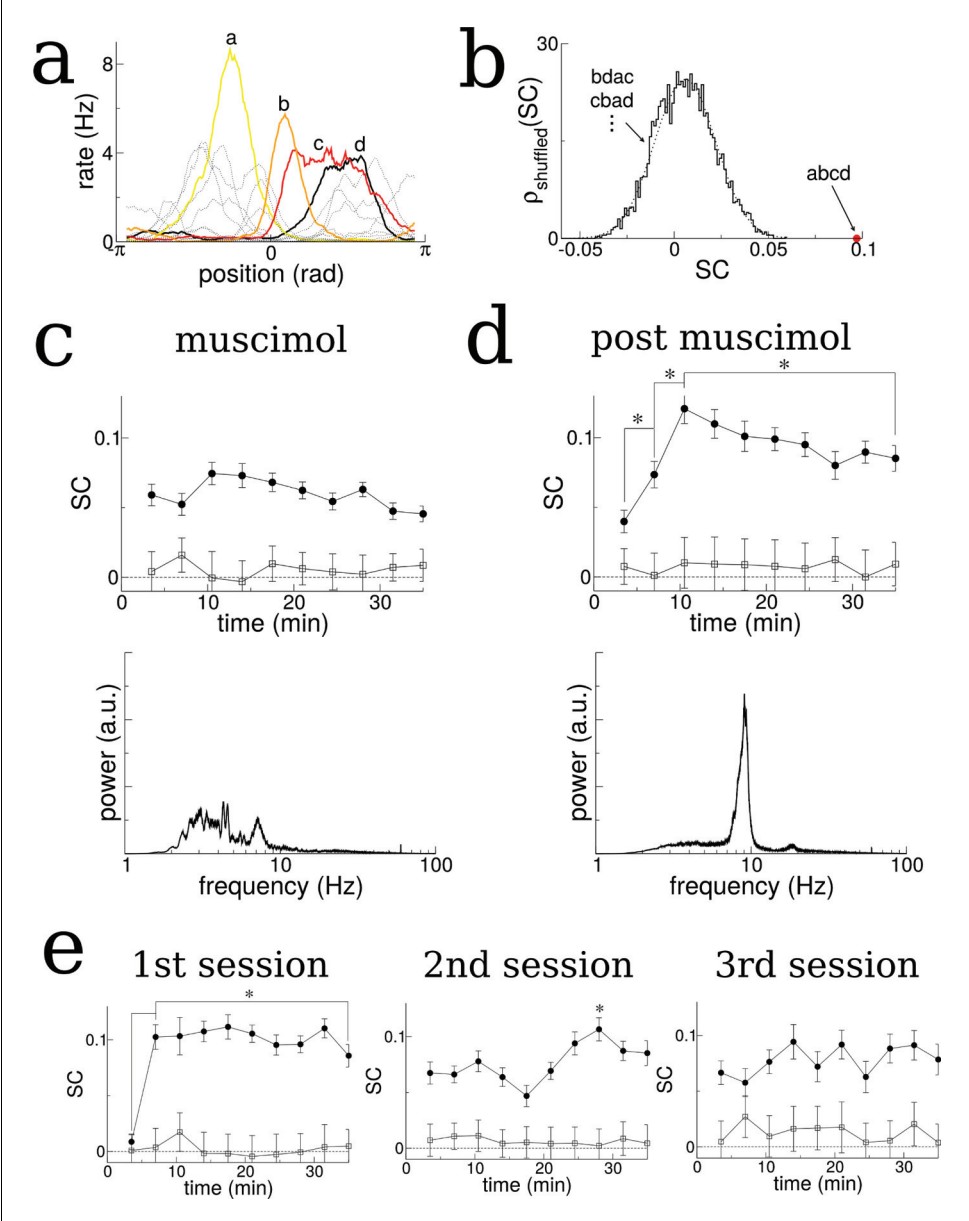

**Figure 7.** The SC of place cells in rat hippocampus during the exploration of a novel track shows an initial increase and plateau, but only when theta is present. (a) Sample firing rate profiles from place cells recorded from CA1 of rat hippocampus during exploration of a novel periodic track with an illustration of how cells can be ordered according to their phases. (b) The SC calculated over the entirety of the experiment (35 min) given the correct ordering (red dot) and for 5000 reshuffled orderings. (c) The time course of SC given the proper ordering (solid symbols) does not show any dynamics when the medial septum is reversibly inactivated with muscimol. Note, however, the clear separation with the shuffled data (open symbols), indicating that place fields are intact even though theta is disrupted. (d) After a rest period the animal is placed back onto the same track; the SC now exhibits a significant increase given the proper ordering (solid symbols) over the first 10 min of exploration and then plateaus. Note that the bottom plots in (c) and (d) show the power spectrum of hippocampal LFP (CA1), indicating a large reduction of theta power in the muscimol condition. (e) When the animal is allowed to explore a novel track and then placed back on the track for a second and third session, there is a significant increase in the SC only during the first session. Error bars are S.E.M.

DOI: https://doi.org/10.7554/eLife.37388.016

The following figure supplements are available for figure 7:

**Figure supplement 1.** Place cell statistics for rat A991.
DOI: https://doi.org/10.7554/eLife.37388.017

**Figure supplement 2.** Same as *Figure 7—figure supplement 1* for rat A992.
DOI: https://doi.org/10.7554/eLife.37388.018

*Figure 7 continued on next page*

*Figure 7 continued*

**Figure supplement 3.** *Figure 7—figure supplement 3* SC and statistics for cells recorded during three sessions exploring a novel hexagonal track, from two rats.

DOI: https://doi.org/10.7554/eLife.37388.020

**Figure supplement 4.** The time-resolved SC for both rats both with medial septum inactivation (muscimol) and after recovery (post muscimol) and for different selection criteria of the tuning index (TI) of place fields.

DOI: https://doi.org/10.7554/eLife.37388.019

correlation was not always significantly different from that calculated from shuffled orderings of the cells (unshaded points in *Figure 7—figure supplement 3a*, rat A991).

## Discussion

### Summary

We have presented a computational model of a hippocampal place-cell network in order to investigate how the exploration of novel environments shapes the patterns of recurrent synaptic connectivity. Because place-fields remap randomly from one environment to the next, the recurrent connectivity, shaped by previous learning, is initially uncorrelated with place-cell activity in a novel environment. Our major finding is that the learning rate during the exploration of a novel environment depends almost entirely on the product of the autocorrelation of place-cell activity and the window for plasticity. The integral of this product determines the growth rate of a global, network-wide pattern of synaptic connectivity, which results in strong local recurrence and long-range competition. It is this mode which drives spontaneous replay activity in our model network. The growth rate of this mode is maximum when place-cell activity is periodically modulated on a time-scale commensurate with the plasticity rule, which for realistic time constants yields frequencies in the theta range. Furthermore, lower and higher frequencies than theta lead to learning rates which are orders of magnitude slower. This suggests that the role of theta is to accelerate learning. Note that the overall rate of plasticity is not affected by the presence of oscillations. The number of spike pairs, and hence the number of potentiations and depressions, depends only on the firing rates. Rather, theta oscillations generate repeated pre-post pairings in both directions, which coupled with a plasticity rule with dominant potentiation at short latencies bias plasticity toward potentiating events for neurons with neighboring place fields. One signature of this mechanism is a continuous increase in the pairwise cross-correlation in the activity of neighboring place-cells leading up to a critical time. We have found evidence consistent with this by analyzing the activity of simultaneously recorded hippocampal place cells in a rat during the exploration of a novel track.

### The assumption of plasticity at recurrent synapses

In our model we have assumed that plasticity occurs only in the recurrent excitatory synaptic connections, and not in the feed-forward inputs. Therefore we also assume that the place-field input, which peaks at the spatial position of the virtual animal at given moment in time, is itself stable. In fact, consistent with this assumption, it seems most place cells are active from the outset of exploration of a new environment, although see (*Hill, 1978*; *Frank et al., 2004*; *Monaco et al., 2014*). Furthermore cells tend to exhibit only subtle changes in the size and shape of their place fields over time (*Mehta et al., 1997*; *Mehta et al., 2000*), also consistent with our model, see *Figure 2—figure supplement 2b*. On the other hand, it has been shown in area CA1 that some place-cells exhibit place-fields only after several minutes of exploration (*Frank et al., 2004*; *Frank et al., 2006*). Recent intracellular recordings indicate that appearance of these 'hidden' place-fields requires the coincidence of active dendritic events and synaptic input via the Schaeffer collaterals (*Lee et al., 2012*; *Bittner et al., 2015*; *Bittner et al., 2017*). It may be that this mechanism for place-cell 'generation' is particularly salient in cells of area CA1 by virtue of being uniquely positioned to compare and integrate both entorhinal and hippocampal inputs. In any case the strongly recurrent nature of the network we study may make it a more relevant model for circuits in area CA3.

Nonetheless we would expect that changes in spiking activity arising in CA3 due to plasticity in the recurrent connectivity, as in our model, would be reflected in similar changes in the spiking

activity of CA1 cells due to the direct inputs via the Schaeffer collaterals. More specifically, in contrast to plasticity mechanisms leading to the formation of place cells themselves, here we have modeled on how plasticity shapes the recurrent connections between already-formed place cells. We find that pairwise correlations between place cells with nearby preferred locations increases during exploration of a novel environment. Assuming such an increase occurs in a strongly recurrent circuit in CA3, we would also expect to observe an increase in correlation in target CA1 pyramidal cells, as long as there exists some mapping from CA3 place cells to CA1 place cells. Such a mapping could be a simple random projection or a more ordered relationship. Recent work suggests that place fields of CA1 pyramidal cells on linear tracks are built up of a weighted sum of CA3 inputs from positions surrounding the relevant place field (*Bittner et al., 2017*); in such a case the increased correlation in CA3 activity would be shared by CA1 output. Indeed, the data we have analyzed from place cells of area CA1 show increase sequential correlation as predicted by our recurrent model. A recent computational model of STDP-induced formation of place fields in CA1 cells via the feed-forward excitatory connections from CA3 (Schaeffer Collaterals) suggests a role for theta in speeding up place cell formation (*D'Albis et al., 2015*). This raises the intriguing suggestion that theta may play a key role both in place-cell formation through plasticity of feed-forward inputs, and also in the emergence of replay through plasticity of recurrent synaptic connections as indicated by our work here.

## Remapping of place fields for different directions of motion on the same track

Hippocampal place cells actually exhibit global remapping of their place fields depending on the direction of motion of the animal on linear tracks (*McNaughton et al., 1983*; *Muller et al., 1994*). This is perhaps not surprising given that the behaviorally relevant information for the animal is not just the position along the track but also the way in which it is facing; for example this determines how far away a potential reward is, if located at one or both ends of the track. Studies using periodic tracks have shown no such global remapping, but rather some degree of rate remapping (*Schwindel et al., 2016*), that is the direction of motion affects the shape and amplitude of place fields, but not their position. In the data we have analyzed there is very weak remapping, see *Figure 7—figure supplements 1c* and *2b*, and so the assumption of invariance of place field to direction of motion is a good one. In our model we have made this assumption. The consequence of this is that while exclusively clockwise (CW) or counter-clockwise (CCW) motion will lead to highly asymmetric recurrent connectivity, exploration of both directions will give rise to much more symmetric connectivity, see *Figure 2a*. In practice any trajectory over a finite amount of time will have a directional bias; in the data we have analyzed the rat spends a slightly larger fraction of the time moving CW compared to CCW, and this will necessarily lead to asymmetries in the connectivity. In linear tracks, due to the global remapping such asymmetries should be even more pronounced.

## Forward versus backward replay

The inevitable asymmetry in the recurrent connectivity of our model place-cell network due to plasticity during exploration strongly biases spontaneous activity. On a periodic track this replay would be exclusively CW or CCW depending on the corresponding bias in motion during exploration, while learning on a linear track would always produce forward replay. Previous work has shown that perfectly symmetric connectivity can give rise to both forward and backward replay in equal measures, due to spontaneous symmetry breaking of activity (*Romani and Tsodyks, 2015*). We would argue that such symmetric connectivity is not robust for the reasons given above, although we cannot rule out the existence of homeostatic mechanisms which would conspire to make it so. Rather, we propose here an alternative mechanism for generating backward replay given asymmetric connectivity, based on local sensory input.

Specifically, if global input to our model network is homogeneous then replay occurs only in one direction. This is due to the asymmetry in the connectivity which comes about through a bias in the direction of motion of the animal during exploration. On the other hand, if the input is not homogeneous, but rather spatially localized, then the replay dynamics can be qualitatively distinct. This difference is mediated by the short-term synaptic depression mechanism present in the model. With a strong, spatially localized input, synapses to downstream neurons (in the sense of the asymmetric bias of the connectivity) become rapidly depressed, see *Figure 2—figure supplement 6*. This forces

the activity to travel backward with respect to the bias in the connectivity. In fact, in experiment, when local spatial input is absent, for example when the animal is sleeping in a rest box, forward replay is predominant (*Roumis and Frank, 2015*). On the other hand, both backward and forward replay are observed when the animal is awake but quiescent on a given track. This is precisely when locally sensory cues are available to the animal, and could potentially shape spontaneous replay events. Recent work shows that some neurons in area CA2 fire more strongly during awake quiescence than during exploration (*Kay et al., 2016*); they may be providing information regarding local sensory cues. In our scenario the mechanisms leading to forward versus backward replay are distinct and therefore in principle relevant replay statistics such as replay velocity and spiking intensity should also be different.

## Robustness to changes in the plasticity model and to the presence of spike correlations

Here we have considered a simple phenomenological model of plasticity which depends on the timing of spike pairs. Taking into account spike triplets as opposed to only pairs does not alter our findings qualitatively, see *Figure 5e*, although we are necessarily ensuring a balanced rule through our choice of parameters. Fits of heuristic spike timing-dependent models to data from slice experiments yield parameter values which do not lead to balanced rules such as the ones we have used here (*Pfister and Gerstner, 2006*). Specifically, when the repetition-rate of spike pairs is high, in-vitro STDP is dominantly potentiating; for such a rule there would be no non-monotonic dependence of the learning rate on modulation frequency at high rates. The mechanism we propose for theta would not be operative in that case. However, in a recurrent network model, unbalanced plasticity rules would quickly lead to synapses either saturating to maximum efficacy or vanishing completely; non-trivial network structure is therefore not possible with such rules. One solution to this problem is to complement the heuristic, unbalanced Hebbian rule, extracted from slice experiments, with a non-Hebbian heterosynaptic rule, as well as additional, slower homeostatic mechanisms (*Zenke et al., 2015*). It is unclear if the learning rate of this rule would also exhibit a non-monotonic dependence of the modulation frequency, as in our rule. However, heuristic rules describing synaptic plasticity in-vivo are still largely unknown. Therefore one reasonable approach to studying plasticity in recurrent networks in-vivo is to choose the simplest possible rule which allows for the emergence of non-trivial structure; this has been our approach here. It remains to be studied how more realistic voltage- or calcium-based plasticity rules interact with the theta-modulation to affect learning in recurrent networks (*Clopath et al., 2010*; *Graupner and Brunel, 2012*), although at a single-synapse one can find qualitatively similar regimes for an array of plasticity rules in the presence of pre- and post-synaptic oscillations (*Albers et al., 2013*).

Our results clearly do not depend on the actual spike timing since our model neurons generate spikes as Poisson processes; rather, all lasting changes in the connectivity are due to time-varying modulations of the firing rates. In fact, recent work with a spiking neuron model suggests that such modulations in the firing rate, as opposed to exact spike timing, are sufficient to explain the effect of plasticity from STDP and more realistic calcium-based plasticity rules in general (*Graupner et al., 2016*). In any case the contribution of pairwise spike correlations to the evolution of the recurrent connectivity can be formally taken into account in (*Equation 4*), that is via its affect on the auto-correlation of place-cell activity.

## Theta sequences and phase precession

The increase in SC during theta activity in our model is due to the emergence of theta sequences, *Figure 2—figure supplement 5*, a consequence of the strengthening of the recurrent excitatory connectivity during exploration. The emergence of theta sequences also gives rise to phase precession of the place cell activity. Specifically, the peak of the place cell firing rate shifts to earlier phases of the theta rhythm as the animal enters the place field. This mechanism was first studied in (*Tsodyks et al., 1996*), in which theta sequences arose through asymmetry in the recurrent connectivity. A very similar effect can be achieved through short-term synaptic depression even without the need for asymmetric connectivity (*Romani and Tsodyks, 2015*). The reason is that the motion of the animal, and hence the sequential activation of place cells, generates an asymmetric pattern of synaptic depression (upstream synapses are more depressed than downstream ones). This is the

mechanism responsible for theta sequences and phase precession in our model. Phase precession therefore emerges over time in our model, and is not present during early exploration. We have also studied the effect of phase precession on learning numerically by generating theta sequences directly via the place-field input itself (not shown). Preliminary simulations suggest that phase precession can actually speed up learning, although the mechanism is non-trivial and requires additional study.

## Other network models of place-cell activity

Recurrent network models for place-cell activity provide a parsimonious explanation for electro-physiological phenomena associated with exploratory behavior as well as the generation of sharp-wave bursts during awake quiescence and sleep (*Romani and Tsodyks, 2015*; *Tsodyks et al., 1996*; *Shen and McNaughton, 1996*; *Cutsuridis and Hasselmo, 2011*; *Jahnke et al., 2015*). In recent theoretical work (*Romani and Tsodyks, 2015*) sharp-wave-like bursts were generated spontaneously by virtue of the spatial modulation of the recurrent connectivity, which drives an instability to traveling waves in the absence of place-field input. The presence of short-term depression modulates the amplitude of the waves, leading to narrow bursts. This is the same mechanism we have used here. Alternatively, recent work with a biophysically detailed spiking network model focused on the role of nonlinear dendritic integration on the generation of replay during SWRs (*Jahnke et al., 2015*). In that work the authors found that the exploration of a virtual linear track in the presence of pairwise STDP led to highly asymmetric connectivity; this could generate replay activity given a sufficiently synchronous external input which recruited nonlinear dendritic events. In our work, we have sought to explain the replay as an emergent phenomenon which depends only on the network-wide organization of synaptic structure. In doing so we have considered a simple stochastic firing rate model which allowed us to fully characterize how interplay between the plasticity rule and the place-cell activity affects learning. Nonetheless, a detailed reproduction of the phenomenology of SWRs certainly requires mechanisms we have not included here. In particular, while our model generates sharp-wave-like bursts, it does not generate high-frequency ripples, which are likely generated by networks of inhibitory interneurons in CA1 (*Buzsáki, 2006*).

## Spatial learning

It seems reasonable that the learning of tasks which depend on spatial information require the formation of an internal representation of the relevant environment. This is the process we have studied here. While we have not modeled any particular cognitive task, we propose that the network-wide organization of synaptic structure, in order that it be in concordance with the place-field distribution of place cells, should be a necessary step in spatial learning tasks. Our results suggest that this process is dramatically sped up by modulating place-cell activity in the theta range, which is one possible role of this prominent rhythm. Interestingly, while theta modulation is prominent in land-going rodents, it is absent in the bat (*Yartsev and Ulanovsky, 2013*). Although we cannot purport to know why this is, we find that it actually may be consistent with the theoretical mechanism for learning rates we put forth here. Namely, to attain a high learning rate the AC of place-cell activity should be modulated on a time-scale commensurate with the window for plasticity. In the case of land mammals like rats and mice, the behavioral time-scale related to spatial navigation may be too slow, therefore necessitating oscillatory modulation in the form of theta. In the case of flying animals, like the bat, changes in sensory input alone are already likely to occur on the order of tens to hundreds of ms due to the high velocity of flight, thereby obviating the need for theta. This speculation is based on the modulatory effect of ongoing oscillations on learning in our computational model, and is therefore not yet supported by experimental evidence. Indeed, our theoretical prediction is that more generally, for learning to occur on behaviorally relevant time scales, neuronal activity should vary somehow on a time-scale commensurate with synaptic plasticity mechanisms. One means of achieving this is to have a broad window for synaptic plasticity, on the order of seconds, so-called behavioral time-scale plasticity (*Bittner et al., 2017*). Alternatively, internally generated rhythms may serve to modulate neuronal activity on non-behavioral time-scales, with a similar effect.

## Materials and methods

### Network Model

We simulate a network of $N$ place-cells. The firing rate of neuron $i$ is given by

$$\tau \dot{r}_i = -r_i + \phi\left(\frac{1}{N}\sum_{j=1}^{N}\tilde{w}_{ij}x_jr_j + I_i(t)\right),$$ (8)

where the time constant $\tau = 10$ ms and $N = 100$ for all simulations except for those in *Figure 6* in which $N = 200$. We take $\phi(I) = \alpha\ln\left(1 + e^{I/\alpha}\right)$ with $\alpha = 1$ Hz for all simulations. The input

$$I_i = I_0 + I_{\mathrm{PF}}\cos(\theta_i^\mu - x(t))(1 + I_{\mathrm{theta}}\cos(2\pi ft)),$$ (9)

where $\theta_i^\mu$ is the center of the place field for cell $i$ on track $\mu$. These place field positions are random and uncorrelated for each cell $i$ on different tracks. The position of the animal is given by $X(t)$. For the simulations in all figures except *Figure 2—figure supplement 2* the position was calculated by assuming that the velocity of the animal was random with a characteristic time constant of 10 sec. Specifically the velocity is given by $v = v_0 + v_1(t)$ where the mean velocity $v_0 = 0.5$ rad/sec and

$$\tau_v \dot{v}_1 = -v_1 + \sigma_v\xi_v(t),$$ (10)

where $\xi_v$ is a Gaussian white-noise process with mean zero and unit variance, with $\tau_v = 10$ sec and $\sigma_v = 2$ rad/sec. The position of the animal is then given by solving $\dot{X} = v$. For *Figure 2—figure supplement 2* the velocity is taken to be a constant $v = 1$ rad/sec and the position $X = vt$.

The short-term synaptic depression variable $x_i$ obeys

$$\dot{x}_i = \frac{(1 - x_i)}{\tau_x} - U_0 x_i r_i,$$ (11)

with $\tau_x = 800$ ms and $U_0 = 0.0008$.

### Synaptic weights

The synaptic weight from cell $j$ to cell $i$ is $\tilde{w}_{ij} = w_{ij} - w_I$, where $w_I$ represents a global inhibitory feedback. The weights $w_{ij}$ are plastic and evolve according to a spike-timing dependent plasticity rule. In order to generate spike trains we take the firing rates of the cells as the underlying rates for a Poisson process. Specifically, the probability that a cell $i$ produces a spike in a given time step $\Delta t \ll 1$ is defined as $Pr(\mathrm{spike}) = r_i(t)\Delta t$.

We consider a simple pairwise STDP rule (*Kempter et al., 1999*). To implement this numerically, if we take the point of view of a neuron $i$, which receives synaptic input from a neuron $j$, then the weight $w_{ij}$ will change value for each pre-post spike pair. Specifically, the weight is strengthened every time there is a post-synaptic spike. The amount it is strengthened by depends on the how long ago the pre-synaptic spike occurred. If it is discounted exponentially then we need only take into account a presynaptic variable $r_j$, where

$$\dot{r}_j = -\frac{r_j}{\tau_+} + S_j(t),$$ (12)

and $S_j(t) = \sum_k \delta\left(t - t_j^k\right)$ is the train of spikes emitted by neuron $j$. Then, when the postsynaptic spike occurs at time $t$ we increase $w_{ij} \rightarrow w_{ij} + A_+ r_j(t)$. Similarly, $w_{ij}$ is weakened whenever there is a presynaptic spike by an amount which depends on the spike times of the postsynaptic neuron $i$. Therefore, we can keep track of a postsynaptic variable $o_i$, where

$$\dot{o}_i = -\frac{o_i}{\tau_-} + S_i(t),$$ (13)

and everytime there is a presynaptic spike at time t we update $w_{ij} \rightarrow w_{ij} - A_- o_i(t)$. Weights are bounded below by zero and above by $w_{max}$. This fully characterizes the plasticity rule. The parameters $A_+ = 0.1$, $\tau_+ = 20$ ms, $A_- = 0.1/3$, $\tau_- = 60$ ms unless otherwise stated.

## Initial Condition

The weight matrix $w_{ij}$ was trained by simulating exploration on 10 distinct linear tracks for one hour each. The value of $w_{ij}$ was taken as a constant 40 for all synapses at the beginning of exploration of the first environment and the maximum possible weight was set at $w_{max} = 80$. See *Figure 2—figure supplement 1* for statistics of the synaptic weights during the training process. For the connectivity profiles, we calculate the mean synaptic weight between pairs of neurons with a difference in place field location $\Delta\theta$ at the given time of the snapshot. There are no autapses, but the curve is made continuous at $\Delta\theta$ by interpolating between adjacent points. The connectivity is normalized by subtracting the mean and dividing by 40.

## Details for *Figure 2*

To generate the figure in *Figure 2b* we calculate the SC in 1 s bins during the entire simulation. Every 180 s there is a 3 s period during which the external theta-modulated place field input is removed, in order to model awake quiescence. Activity during this period is spontaneous and is considered to be bursts (black circles). During burst activity we only calculate the SC for the second and third seconds because it takes some time for the place field activity to die away and the burst activity to emerge, for example SC for bursts is calculated for seconds 181–182 and 182–183 but not for 180–181. For the simulation in *Figure 2b* the place field input is simply removed when the animal stops, that is the external input is set to a constant value of $I_0 = 3$ Hz while for *Figure 2c* it is set to $I_0 = -0.75$ Hz. Changing $I_0$ does not significantly affect the value of SC, only the degree of burstiness of the spontaneous activity, see *Figure 2—figure supplement 7*. The inset in *Figure 2b* was generated by binning the SC of the total network activity into 3 min bins. The 'LFP' in *Figure 2c* is the network-averaged input current to the neurons, that is the network-average of the argument in the firing rate equations. Additional parameters are: $I_0 = 3$ Hz, $I_{PF} = 25$ Hz, $I_{theta} = 1$, $f = 8$ Hz, $w_I = 65$.

## Details for *Figure 3*

The curves shown in *Figure 3a* are a cartoon meant to illustrate how the recurrent connectivity can be decomposed into a spatial Fourier series which include even (cosine) and odd (sine) terms. The amplitude of an even term (its coefficient) can lead to a transition in the network dynamics when it reaches a critical value. In *Figure 3c* the coefficients $w_{even}$ and $w_{odd}$ are the first cosine and sine Fourier coefficients of the mean recurrent connectivity are calculated as in *Figure 2* at a given time during the simulation. Parameters in (c) are the same as in *Figure 2*.

## Details for *Figure 4*

The parameters in (a) and (b) are the same as in *Figure 2*. Parameters in (c) and (d) are the same as in *Figure 2* with the sole exception of the STDP rule. For the anti-symmetric case (c) the parameters are $A_+ = 0.1$, $\tau_+ = 40$ ms, $A_- = 0.1$, $\tau_- = 40$ ms while for (d) they are $A_- = 0.1$, $\tau_+ = 20$ ms, $A_+ = 0.1/3$, $\tau_- = 60$ ms.

## Details for *Figure 6*

The virtual animal has explored thirty distinct environments for one hour each. In each environment the coding level is one half, that is one half of the neurons are modeled as place cells (randomly assigned place field location from uniform distribution around the track) and the other half receive only constant background input with $I_0 = 0$ Hz. Plasticity occurs via STDP as before between all cell pairs. Place cells in any given environment are chosen randomly with equal probability; hence the overlap in place-cell representation between any two environments is on average one half. The number of neurons is N = 200, so that in any environment there are still 100 place cells, as in previous simulations.

## Derivation of continuous learning rule from pairwise and triplet STDP rule

### Pairwise rule

We can write down an approximate description for the evolution of $w_{ij}$ which is accurate if the STDP occurs much more slowly than changes in the firing rates (see Kempter et al. PRE 1999 for details). For the case of inhomogeneous Poisson processes the equation is

$$\dot{w}_{ij} = -A_- \int_{-\infty}^{0} dT \cdot e^{T/\tau_-} r_j(t) r_i(t+T) + A_+ \int_{0}^{\infty} dT \cdot e^{-T/\tau_+} r_j(t-T) r_i(t). \tag{14}$$

As we are only interested in the evolution of the weights on a long time-scale (longer than the time-scale of fluctuations in the rates) we will consider the time averaged equation

$$\langle \dot{w}_{ij} \rangle_t = -A_- \int_{-\infty}^{0} dT \cdot e^{T/\tau_-} \langle r_j(t) r_i(t+T) \rangle_t + A_+ \int_{0}^{\infty} dT \cdot e^{-T/\tau_+} \langle r_j(t-T) r_i(t)_t \rangle. \tag{15}$$

For a stationary process we have $\langle r_j(t-T) r_i(t) \rangle_t = \langle r_j(t) r_i(t+T) \rangle_t$. Therefore, as long as we restrict our analysis to stationary processes (which is the case here), we can write (*Equation 15*) in the compact form

$$\dot{w}_{ij} = \int_{-\infty}^{\infty} dT \cdot K(T) r_j(t) r_i(t+T),$$

$$K(T) = \begin{cases} A_+ e^{-T/\tau_+} & \text{if } T \geq 0 \\ -A_- e^{T/\tau_-} & \text{if } T < 0. \end{cases} \tag{16}$$

If neurons are arranged along a periodic track and can be parameterized according to their position (phase) $\theta$, then (*Equation 16*) becomes (*Equation 3*). Note that (*Equation 3*) is only used for analysis. Plasticity in the numerical simulations is modeled as described in the preceding section.

## Triplet rule

The STDP rule which depends only on spike pairs cannot account for some general experimental findings, including the dependence of LTP on presentation frequency. A triplet rule can describe these effects (*Pfister and Gerstner, 2006*). The triplet rule is implemented as above, but by adding one additional variable to keep track of per neuron. Specifically, we rename the previous pre-synaptic and post-synaptic variables $o_1$ and $r_1$ and add two more $o_2$ and $r_2$. Now if there is a spike in neuron $i$ at time $t$, then the weight $w_{ij}$ is potentiated by an amount $r_{1,j}\left(A_2^+ + A_3^+ o_{2,i}(t - \epsilon)\right)$, where the $\epsilon$ means that you take the value of the post-synaptic variable $o_2$ before it is incremented due to the spike. Similarly, the weight $w_{ji}$ is depressed by an amount $o_{1,j}(t)\left(A_2^- + A_3^- r_{2,i}(t - \epsilon)\right)$.

One can write an equation similar to (*Equation 3*) for the triplet rule, which is exact if the learning is slow compared to the rate dynamics. The full equation, with pairwise and triplet interactions is

$$\begin{aligned} \dot{w}_{ij} &= \int_{-\infty}^{\infty} d\tau K_{11}(\tau) r_i(t) r_j(t+\tau) \\ &- A_3^- \int_{0}^{\infty} d\tau_1 \int_{0}^{\infty} d\tau_2 e^{-\tau_1/\tau_-} e^{-\tau_2/\tau_x} r_i(t) r_j(t-\tau_1) r_i(t-\tau_2) \\ &+ A_3^+ \int_{0}^{\infty} d\tau_1 \int_{0}^{\infty} d\tau_2 e^{-\tau_1/\tau_+} e^{-\tau_2/\tau_y} r_j(t) r_i(t-\tau_1) r_j(t-\tau_2). \end{aligned} \tag{17}$$

The kernel $K_{11}$ is just the standard pairwise one from before, where we write 11 to mean 'one pre-synaptic spike and one postsynaptic spike'. The kernel $K_{12}$ is the one for 'two presynaptic spikes and one postsynaptic spike'. The STDP rule is multiplicative, that is the potentiation due to the triplet is the product of $r_1$ and $o_2$ which implies that $K_{21}(\tau_1, \tau_2) = K_{11}(\tau_1)\tilde{K}_{21}(\tau_2)$, where $\tau_1 = t_{pre} - t_{post,2}$ and $\tau_2 = t_{post,1} - t_{post,2}$, so $\tau_1 \leq 0$ and $\tau_2 \leq 0$, and we are only integrating over the potentiating part of $W_{11}$ which is also clear from the STDP rule.

## Plasticity in network of place cells on periodic track

We first consider a simple case where the place cell activity is given by $r(\theta, t) = r_0 + r_{PF} \cos(\theta - vt)$, that is the rate dynamics trivially follow the motion of the virtual animal and are not 'theta'-modulated. Also, we are taking a continuum limit in which the index of a neuron $i$ is replaced with the position of its place field along the track $\theta$. In (*Equation 3*) we have the product

$$
\begin{aligned}
r(\theta,t)r(\theta',t+T) &= r_0^2 + r_{PF}^2 \cos(\theta - vt)\cos(\theta' - vt - vT) + f(t), \\
&= r_0^2 + r_{PF}^2 \cos(\theta - \theta' + vT) + f(t), \\
&= r_0^2 + \tfrac{r_{PF}^2}{2}\left(\cos(\theta - \theta')\cos(vT) + \sin(\theta - \theta')\sin(vT)\right) \\
&\quad + f(t),
\end{aligned}
\tag{18}
$$

where $f(t)$ is a time-periodic function. The first two terms in (*Equation 18*) will therefore determine the mean growth of the even and odd modes of the connectivity on a slow time-scale, while the remainder will lead to fluctuations about this mean on a faster time scale. (This time-scale separation will be conducted more formally in a later section) Averaging over the fast time eliminates these fluctuations. Therefore the rate of change of the connectivity can be expressed as

$$
\langle \dot{w} \rangle_t (\theta - \theta') = \dot{w}_0 + \dot{w}_{\text{even}}\cos(\theta - \theta') + \dot{w}_{\text{odd}}\sin(\theta - \theta'),
\tag{19}
$$

where the brackets on the l.h.s. indicate that we have taken a time-average to eliminate time-oscillating terms. Therefore, (*Equation 3*) reduces to

$$
\dot{w}_0 = r_0^2 \int_{-\infty}^{\infty} dT \cdot K(T),
$$

$$
\dot{w}_{\text{even}} = \frac{r_{PF}^2}{2} \int_{-\infty}^{\infty} dT \cdot K(T)\cos(vT),
\tag{20}
$$

$$
\dot{w}_{\text{odd}} = \frac{r_{PF}^2}{2} \int_{-\infty}^{\infty} dT \cdot K(T)\sin(vT).
\tag{21}
$$

Performing these integrals gives

$$
\dot{w}_0 = (A_+ \tau_+ - A_- \tau_-) r_0^2,
\tag{22}
$$

$$
\dot{w}_{\text{even}} = \frac{r_{PF}^2}{2}\left[ \frac{A_+ \tau_+}{1 + \tau_+^2 v^2} - \frac{A_- \tau_-}{1 + \tau_-^2 v^2} \right],
\tag{23}
$$

$$
\dot{w}_{\text{odd}} = \frac{r_{PF}^2}{2}\left[ \frac{A_+ \tau_+^2 v}{1 + \tau_+^2 v^2} + \frac{A_- \tau_-^2 v}{1 + \tau_-^2 v^2} \right].
\tag{24}
$$

We see from (*Equation 22*) that the balance condition $A_+ \tau_+ = A_- \tau_-$ must hold to avoid the synapses from all reaching their maximum or minimum value. At this point we would also note that for any reasonable velocity the growth rate of the even mode is extremely small and far from the maximal possible growth rate. Specifically, the velocity has unit of radians. For a three meter track, a rat running at 50 cm/s would have a velocity of about 1 rad/sec. Typical time scale for STDP are 50 ms or 0.05 s. Therefore the non-dimensional quantity $\tau^2 v^2 = 0.0025 \ll 1$ and, given the balance condition, $\dot{w}_{\text{even}} \sim 0$.

Including the triplet interactions by using (*Equation 17*) gives

$$
\begin{aligned}
\dot{w}_0 &= (A_+ \tau_+ - A_- \tau_-) r_0^2 + A_3^+ \left[ \tau_+ \tau_y r_0^3 + \frac{\tau_+ \tau_y r_0 r_{PF}^2}{2(1 + \tau_y^2 v^2)} \right] \\
&\quad - A_3^- \left[ \tau_- \tau_x r_0^3 + \frac{\tau_- \tau_x r_0 r_{PF}^2}{2(1 + \tau_x^2 v^2)} \right],
\end{aligned}
\tag{25}
$$

$$
\begin{aligned}
\dot{w}_{\text{even}} &= \frac{r_{PF}^2}{2}\left[ \frac{A_+ \tau_+}{1 + \tau_+^2 v^2} - \frac{A_- \tau_-}{1 + \tau_-^2 v^2} \right] + A_3^+ \frac{\tau_+ \tau_y r_0 r_{PF}^2}{2(1 + \tau_+^2 v^2)}\left[ 1 + \frac{(1 + \tau_+ \tau_y v^2)}{(1 + \tau_y^2 v^2)} \right] \\
&\quad - A_3^- \frac{\tau_- \tau_x r_0 r_{PF}^2}{2(1 + \tau_-^2 v^2)}\left[ 1 + \frac{(1 + \tau_- \tau_x v^2)}{(1 + \tau_x^2 v^2)} \right],
\end{aligned}
\tag{26}
$$

$$\dot{w}_{\text{odd}} = \frac{r_{PF}^2}{2}\left[\frac{A_+\tau_+^2 v}{1+\tau_+^2 v^2} + \frac{A_-\tau_-^2 v}{1+\tau_-^2 v^2}\right] + A_3^+\frac{\tau_+\tau_y vr_0 r_{PF}^2}{2(1+\tau_+^2 v^2)}\left[\tau_+ + \frac{(\tau_+ - \tau_y)}{1+\tau_y^2 v^2}\right]$$
$$+ A_3^-\frac{\tau_-\tau_x vr_0 r_{PF}^2}{2(1+\tau_-^2 v^2)}\left[\tau_- + \frac{(\tau_- - \tau_x)}{1+\tau_x^2 v^2}\right] \tag{27}$$

From (*Equation 25*) we see there are two additional balance conditions to avoid saturation or decay to zero of synaptic weights: $\tau_x = \tau_y$ and $A_3^+\tau_+\tau_y = A_3^-\tau_-\tau_x$. Once we apply the balance conditions the growth rate of the mean connectivity $w_0$ is identically zero.

## Including periodic modulation

Here we consider the case where the place cell activity is modulated periodically with a frequency $f$, that is $r(t) = r_0 + r_{\text{PF}}\cos(\theta - vt)(1 + r_{\text{theta}}\cos(2\pi ft))$. This can be rewritten as

$$r(t) = r_0 + r_{\text{PF}}\cos(\theta - vt) + \frac{r_{\text{PF}}r_{\text{theta}}}{2}(\cos(\theta - v_+t) + \cos(\theta - v_-t)), \tag{28}$$

where $v_+ = v + 2\pi f$ and $v_- = v - 2\pi f$. Now when we plug (*Equation 28*) into (*Equation 3*) or (*Equation 17*) we find that each of the three cosine terms makes an independent contribution. The reason is that the cross terms (which arise due to the product of rates) lead to time periodic terms which vanish once we average the equation. Therefore we end up with equations of the form (*Equation 22–24*) or (*Equations 25–27*) but with three times the number of terms, that is for $v$, $v_+$ and $v_-$. These equations can, however, be simplified by considering the limit where the ratio $\epsilon = \frac{v}{2\pi f} \ll 1$. For example, when $f = 8$ Hz and $v = 1$ rad/sec (50 cm/sec on a three meter track) then $\frac{v}{2\pi f} \sim 0.02$. Slower running speeds or faster oscillations will make the following approximation even more accurate. In this limit we have (for the pairwise rule)

$$\dot{w}_{\text{even}} = \frac{r_{\text{PF}}^2 A_+\tau_+}{2}\left[\frac{1}{1+\tau_+^2 v^2} - \frac{1}{1+\tau_-^2 v^2}\right]$$
$$+ r_{\text{PF}}^2 r_{\text{theta}}^2 A_+\tau_+\left[\frac{1}{1+4\pi^2\tau_+^2 f^2} - \frac{1}{1+4\pi^2\tau_-^2 f^2}\right] + \mathcal{O}(\epsilon^2), \tag{29}$$

$$\dot{w}_{\text{odd}} = \frac{r_{\text{PF}}^2 A_+\tau_+}{2}\left[\frac{\tau_+ v}{1+\tau_+^2 v^2} + \frac{\tau_- v}{1+\tau_-^2 v^2}\right] + \mathcal{O}(\epsilon). \tag{30}$$

Furthermore, given that $\tau v$ is also a small parameter, we can neglect the first term in (*Equation 29*), which means that the growth rate of the even mode is entirely dominated by the modulation frequency to leading order. Because $f$ could take on any value the $\dot{w}_{\text{even}}$ need not be close to zero. In fact, there is a critical value of f which maximizes the growth rate $f = \frac{1}{2\pi\sqrt{\tau_+\tau_-}}$. On the other hand the growth of the odd mode is not dependent on the modulation frequency to leading order, but only on the motion of the animal.

## Coupled rate dynamics and synaptic plasticity

Now we return to the full problem, namely, (*Equations 1–3*) and we take $\phi$ to be linear for simplicity. In principle (*Equations 1–3*) are extremely challenging to analyze by virtue of the nonlinear combinations of the rates and the connectivity, both of which evolve in time. However, if we assume that the connectivity evolves much more slowly than the rates, the problem simplifies considerably. Specifically, we assume that the changes to the synaptic weights due to each spike pair are small. Formally, we can introduce a small parameter $\epsilon$ and write $A_+ = \epsilon\tilde{A}_+$ and $A_- = \epsilon\tilde{A}_-$, and so $K(T) = \epsilon\tilde{K}(T)$, where all the 'tilded' quantities are order one. In the limit $\epsilon \to 0$ we would then find that (*Equation 1*) is unchanged while (*Equation 3*) becomes $\dot{w}(\theta - \theta') = 0$, that is the connectivity is a constant. This clearly is an approximation since the connectivity will change over time, just slowly. In fact it is precisely this slow evolution that we would like to describe. In order to do this we can define a new, slow time $\tau_s = \epsilon t$ and allow both the firing rate and the connectivity to evolve on both the fast and the slow time scales, which are now taken to be independent. This is known as a multi-scale approach.

Formally we make the following ansatz

$$w = w_0(\theta - \theta', t_s, t) + \epsilon w_1(\theta - \theta', t_s, t) + \vartheta(\epsilon^2), \tag{31}$$

$$r = r_0(\theta, t_s, t) + \epsilon r_1(\theta, t_s, t) + \vartheta(\epsilon^2), \tag{32}$$

where $t_s = \epsilon t$. Therefore, the temporal derivatives in (*Equations 1–3*) can be written $\dot{r} = \partial_t r + \epsilon \partial_{t_s} r$ by using the chain rule. This leads to the system of equations

$$\tau(\partial_t + \epsilon \partial_{t_s})(r_0 + \epsilon r_1 + \ldots) = -r_0 - \epsilon r_1 - \ldots + \frac{1}{2\pi} \int_{-\pi}^{\pi} d\theta'(w_0 + \epsilon w_1 + \ldots)$$
$$(r_0 + \epsilon r_1 + \ldots) + I(\theta, t), \tag{33}$$

$$(\partial_t + \epsilon \partial_{t_s})(w_0 + \epsilon w_1) = \epsilon \int_{-\infty}^{\infty} dT \tilde{K}(T)(r_0(\theta, t) + \epsilon r_1(\theta, t) + \ldots),$$
$$(r_0(\theta', t+T) + \epsilon r_1(\theta', t+T) + \ldots) \tag{34}$$

To leading order we have the equations

$$\tau \partial_t r_0 = -r_0 + \frac{1}{2\pi} \int_{-\pi}^{\pi} d\theta' w_0(\theta - \theta') r_0(\theta') + I(\theta, t), \tag{35}$$

$$\partial_t w_0(\theta - \theta') = 0. \tag{36}$$

(*Equation 36*) shows that, to leading order, the connectivity is independent of the fast time $t$, that is $w_0 = w_0(\theta - \theta', t_s)$. Given this, the term $w_0$ can be treated as a constant in (*Equation 35*), and the rate is given by $r_0 = r_0(\theta, t_s, t)$. At order $\vartheta(\epsilon)$ the equations are

$$\tau \partial_{t_s} r_0 + \tau \partial_t r_1 = -r_1 + \frac{1}{2\pi} \int_{-\pi}^{\pi} d\theta'(w_1 r_0 + w_0 r_1), \tag{37}$$

$$\partial_{t_s} w_0 + \partial_t w_1 = \int_{-\infty}^{\infty} dT \tilde{K}(T) r_0(\theta, t_s, t) r_0(\theta', t_s, t+T). \tag{38}$$

The solution to (*Equation 37*), $r_1$ gives the next order correction to the firing rates. On the other hand, (*Equation 38*) would seem to require solving for both $w_0$ and $w_1$ simultaneously. However, we note that the product of rates $r_0(\theta, t_s, t) r_0(\theta', t_s, t+T)$ can actually be written as $g(\theta - \theta', t_s; T) + f(\theta - \theta', t_s, t; T)$. That is, there is a part which is independent of time, except for the dependence of the slow time through the connectivity, and a part which varies on a regular, non-slow time scale. This fast, time-varying part, has zero mean (it is periodic here) and hence can be eliminated by averaging on an appropriate intermediate time scale, which is still much faster than the slow time scale, that is $\langle f \rangle_t = 0$. Therefore, we can define $w_1$ as that component of the connectivity which is driven by $f$, and hence also will have zero mean and will vanish upon averaging. Finally, combining the order one equation for the rate and the averaged order epsilon equation for the connectivity yields

$$\tau \partial_t r_0 = -r_0 + \frac{1}{2\pi} \int_{-\pi}^{\pi} d\theta' w_0(\theta - \theta') r_0(\theta') + I(\theta, t), \tag{39}$$

$$\partial_{t_s} w_0(\theta - \theta') = \int_{-\infty}^{\infty} dT \tilde{K}(T) f(\theta - \theta', t_s; T).. \tag{40}$$

(*Equations 39 and 40*) are the system we wish to solve here. We first note that the connectivity

can always be decomposed into a Fourier series. Anticipating the fact that the only terms which will not have zero mean in time will be the first cosine and sine terms, we write

$$w_0(\theta - \theta', t_s) = w_0(t_s) + w_{\mathrm{even}}(t_s) \cos(\theta - \theta') + w_{\mathrm{odd}}(t_s) \sin(\theta - \theta'),$$

Given that the forcing term can be written

$$I(\theta, t) = I_0 + I_{\mathrm{PF}} \cos(\theta - vt) + I_{\mathrm{PF}} I_{\mathrm{theta}} \left( \cos(\theta - v_+ t) + \cos(\theta - v_- t) \right). \tag{41}$$

then the firing rates $r_0$ will be

$$
\begin{aligned}
r_0(\theta, t) &= \tilde{r}_0 + r_{even} \cos(\theta - vt) + r_{odd} \sin(\theta - vt) \\
&+ r_{even}^+ \cos(\theta - v_+ t) + r_{odd}^+ \sin(\theta - v_+ t) \\
&+ r_{even}^- \cos(\theta - v_- t) + r_{odd}^- \sin(\theta - v_- t),
\end{aligned} \tag{42}
$$

where, after some linear algebra one finds

$$\tilde{r}_0 = \frac{I_0}{1 - w_0},$$

$$r_{\mathrm{even}} = \frac{I_{\mathrm{PF}}(1 - w_{\mathrm{even}}/2)}{\left(1 - w_{\mathrm{even}}/2\right)^2 + \left(w_{\mathrm{odd}}/2 - \tau v\right)^2},$$

$$r_{\mathrm{odd}} = \frac{I_{\mathrm{PF}}(w_{\mathrm{odd}}/2 - \tau v)}{\left(1 - w_{\mathrm{even}}/2\right)^2 + \left(w_{\mathrm{odd}}/2 - \tau v\right)^2},$$

$$r_{\mathrm{even}}^+ = \frac{I_{\mathrm{PF}} I_{\mathrm{theta}}(1 - w_{\mathrm{even}}/2)/2}{\left(1 - w_{\mathrm{even}}/2\right)^2 + \left(w_{\mathrm{odd}}/2 - \tau v_+\right)^2},$$

$$r_{\mathrm{odd}}^+ = \frac{I_{\mathrm{PF}} I_{\mathrm{theta}}(w_{\mathrm{odd}}/2 - \tau v_+)/2}{\left(1 - w_{\mathrm{even}}/2\right)^2 + \left(w_{\mathrm{odd}}/2 - \tau v_+\right)^2},$$

$$r_{\mathrm{even}}^- = \frac{I_{\mathrm{PF}} I_{\mathrm{theta}}(1 - w_{\mathrm{even}}/2)/2}{\left(1 - w_{\mathrm{even}}/2\right)^2 + \left(w_{\mathrm{odd}}/2 - \tau v_-\right)^2},$$

$$r_{\mathrm{odd}}^- = \frac{I_{\mathrm{PF}} I_{\mathrm{theta}}(w_{\mathrm{odd}}/2 - \tau v_-)/2}{\left(1 - w_{\mathrm{even}}/2\right)^2 + \left(w_{\mathrm{odd}}/2 - \tau v_-\right)^2}.$$

Taking the product of the rates and keeping only those terms with non-zero mean yields the function $f$, which when plugged into (**Equation 40**) gives

$$\dot{w}_0 = (A_+ \tau_+ - A_- \tau_-) r_0^2, \tag{43}$$

$$
\begin{aligned}
\dot{w}_{\mathrm{even}} &= \frac{r_{even}^2 + r_{odd}^2}{2} \left[ \frac{A_+ \tau_+}{1 + \tau_+^2 v^2} - \frac{A_- \tau_-}{1 + \tau_-^2 v^2} \right] \\
&+ \frac{\left(r_{even}^+\right)^2 + \left(r_{odd}^+\right)^2}{2} \left[ \frac{A_+ \tau_+}{1 + \tau_+^2 v_+^2} - \frac{A_- \tau_-}{1 + \tau_-^2 v_+^2} \right] \\
&+ \frac{\left(r_{even}^-\right)^2 + \left(r_{odd}^-\right)^2}{2} \left[ \frac{A_+ \tau_+}{1 + \tau_+^2 v_-^2} - \frac{A_- \tau_-}{1 + \tau_-^2 v_-^2} \right],
\end{aligned} \tag{44}
$$

$$\dot{w}_{\text{odd}} = \frac{r_{\text{even}}^2 + r_{\text{odd}}^2}{2} \left[ \frac{A_+ \tau_+^2 v}{1 + \tau_+^2 v^2} + \frac{A_- \tau_-^2 v}{1 + \tau_-^2 v^2} \right]$$
$$+ \frac{(r_{\text{even}}^+)^2 + (r_{\text{odd}}^+)^2}{2} \left[ \frac{A_+ \tau_+^2 v_+}{1 + \tau_+^2 v_+^2} + \frac{A_- \tau_-^2 v_+}{1 + \tau_-^2 v_+^2} \right] \qquad (45)$$
$$+ \frac{(r_{\text{even}}^-)^2 + (r_{\text{odd}}^-)^2}{2} \left[ \frac{A_+ \tau_+^2 v_-}{1 + \tau_+^2 v_-^2} + \frac{A_- \tau_-^2 v_-}{1 + \tau_-^2 v_-^2} \right].$$

In the limit $v/(2\pi f) \ll 1$ the leading order terms of (*Equations 44 and 45*) are given by (*Equations 6 and 7*).

The solution for the triplet rule can be similarly found by replacing $r_{\text{PF}}$ in (*Equations 25–27*) by $(r_{\text{even}}^2 + r_{\text{odd}}^2)/2$ as well as the analogous terms corresponding to $v_+$ and $v_-$.

## Motion with non-uniform velocity

Up until now we have considered a constant velocity $v$ for ease in calculation. In reality, the motion of the animal will be much more erratic, with both positive and negative speeds during the exploration of the track. In the extreme case of a random walk, in which both clockwise (CW) and counterclockwise (CCW) motion is equally likely, the resulting connectivity is much more symmetric. This occurs because contributions from CW and CCW motion to the even mode simply add, while the same contributions to the odd mode cancel out. We can illustrate this more clearly by considering the continuum learning rule, (*Equation 3*), with a given distribution of velocities. Specifically, we take the simple case where the firing rates can be written $r_0 + r_{\text{PF}} \cos(\theta - vt)$ (we have seen that the solution to the full model can be written in an analogous way to the solution of this simpler problem) and assume a distribution of firing rates $\rho(v)$. From (*Equations 20 and 21*) we have

$$\dot{w}_{\text{even}} = \int_{-\infty}^{\infty} dv \rho(v) \int_{-\infty}^{\infty} dT \cdot K(T) \cos(vT)$$
$$= \int_{-\infty}^{\infty} dT \cdot K(T) \int_{-\infty}^{\infty} dv \rho(v) \cos(vT), \qquad (46)$$

$$\dot{w}_{\text{odd}} = \int_{-\infty}^{\infty} dT \cdot K(T) \int_{-\infty}^{\infty} dv \rho(v) \sin(vT). \qquad (47)$$

If we choose $\rho(v) = \delta(v - v^*)$, that is, a fixed velocity $v^*$, then we recover our results from previous sections. We can allow for a range of velocities by considering a uniform distribution of velocities, that is

$$\rho(v) = \begin{cases} \frac{1}{2v^*}, & |v| \le v^* \\ 0, & |v| > v^*. \end{cases} \qquad (48)$$

For this choice of distribution we find $\dot{w}_{\text{even}} = \int_{-\infty}^{\infty} dT \cdot K(T) \frac{\sin(v^* T)}{v^* T}$, while $\dot{w}_{\text{odd}} = 0$. The function $\sin(x)/x$ is even and decays more slowly than $\cos(x)$ around zero lag, meaning that the AC of the firing rate is qualitatively similar whether $v$ is a constant or distributed. On the other hand, the CC of neurons with disparate place field location will be zero on average if the animal samples both directions equally. Nonetheless, any asymmetry in the velocity distribution (which is expected for any finite length of exploration) will lead to a non-zero odd mode in the connectivity.

## A traveling wave instability of the rate equation for modulated connectivity

In the previous sections we have seen how the connectivity grows in time due to the non-stationarity of the firing rates. In particular, the AC of the place-cell activity drives growth of the even mode, and this growth depends on the STDP rule parameters and the modulation (theta) frequency. Here we show that once the amplitude of the even mode reaches a critical value there is an instability of the network activity in the absence of place field input, which gives rise to spontaneous traveling waves. This is the origin of replay activity in the model.

We consider the firing rate equation in the absence of place field input, that is spontaneous dynamics.

$$\tau \dot{r}(\theta, t) = -r(\theta, t) + \phi\left(\frac{1}{2\pi}\int_{-\pi}^{\pi} d\theta' \, w(\theta - \theta')r(\theta', t) + I_0\right), \tag{49}$$

and a small perturbation about a steady state solution of the form $r(\theta, t) = r_0 + (\delta r_0 + \delta r_{\text{even}}\cos(\theta) + \delta r_{\text{odd}}\sin(\theta))e^{\lambda t}$. Plugging this into (*Equation 49*) leads to a characteristic equation for $\lambda$ the solutions of which are

$$\tau \lambda = -\left(1 - \frac{w_{\text{even}}}{2}\phi'\right) \pm i\frac{w_{\text{odd}}}{2}\phi'. \tag{50}$$

Therefore when $w_{\text{even}} > 2/\phi'$ there is an oscillatory instability with frequency $\omega = \frac{w_{\text{odd}}}{2\tau}\phi'$.

## Including the effects of synaptic depression

A general study of the dynamics in recurrent networks with a ring topology and synaptic depression can be found in (*York and van Rossum, 2009*). Here we outline several calculations which are relevant for our model in the context of plasticity and replay.

### Evolution of recurrent connectivity

Synaptic depression is included in the model in order to generate burst-like activity during awake quiescent periods. The equations are

$$\tau \dot{r} = -r + \phi\left(\frac{1}{2\pi}\int_{-\pi}^{\pi} d\theta' \, w(\theta - \theta')x(\theta', t)r(\theta', t) + I(\theta, t)\right), \tag{51}$$

$$\dot{x} = \frac{1 - x}{\tau_x} - U_0 x r, \tag{52}$$

$$\dot{w}(\theta - \theta') = \int_{-\infty}^{\infty} dT K(T)r(\theta, t)r(\theta', t + T) \tag{53}$$

Here we note that the quadratic nonlinearities $xr$ in (*Equations 51 and 52*) make it impossible to find a closed form solution. However, we also note that the growth rate of the first even mode of the connectivity will still depend only on the product of the STDP window with the autocorrelation (AC) of the firing rate, as before. Therefore, the modulation of the firing rate due to the theta rhythm, the resulting shape of the AC, and its consequence for the growth rate: maximal growth for intermediate frequencies commensurate with the STDP window, is qualitatively the same.

If we assume the place field input is weak, that is $I(\theta, t) = I_0 + I_{\text{PF}}\cos(\theta - vt)$, where $I_{\text{PF}} \ll 1$, then we can linearize the solution. In this case we can write $r(t) = r_0 + r_{\text{even}}\cos(\theta - vt) + r_{\text{odd}}\sin(\theta - vt)$ and $x(t) = x_0 + x_{\text{even}}\cos(\theta - vt) + x_{\text{odd}}\sin(\theta - vt)$, and the product

$$\begin{aligned} xr &= x_0 r_0 + (x_0 r_{\text{even}} + r_0 x_{\text{even}})\cos(\theta - vt) + (x_0 r_{\text{odd}} + r_0 x_{\text{odd}})\sin(\theta - vt), \\ &= \tilde{r}_0 + \tilde{r}_{\text{even}}\cos(\theta - vt) + \tilde{r}_{\text{odd}}\sin(\theta - vt) + h.o.t. \end{aligned} \tag{54}$$

where 'h.o.t.' means higher order terms. Plugging this into (*Equation 52*) gives the solution to x in terms of r

$$x_0 = \frac{1}{1 + \tau_x U_0 r_0}, \tag{55}$$

$$x_{\text{even}} = -U_0 \frac{(v\tau_x r_{\text{odd}} + (1 + \tau_x U_0 r_0)r_{\text{even}})}{(1 + \tau_x U_0 r_0)\left(v^2\tau_x^2 + (1 + \tau_x U_0 r_0)^2\right)}, \tag{56}$$

$$x_{\text{odd}} = -U_0 \frac{(-v\tau_x r_{\text{even}} + (1 + \tau_x U_0 r_0)r_{\text{odd}})}{(1 + \tau_x U_0 r_0)\left(v^2\tau_x^2 + (1 + \tau_x U_0 r_0)^2\right)}. \tag{57}$$

Now we plug the expansion (*Equation 54*) into (*Equation 51*). Doing so yields the following non-linear algebraic equations

$$r_0 = \frac{w_0 r_0}{1 + \tau_x U_0 r_0} + I_0,$$
(58)

$$vr_{\text{even}} = -r_{\text{odd}} + \frac{1}{2}\left(w_{\text{even}}\tilde{r}_{\text{odd}} + w_{\text{odd}}\tilde{r}_{\text{even}}\right),$$
(59)

$$-vr_{\text{odd}} = -r_{\text{even}} + \frac{1}{2}\left(w_{\text{even}}\tilde{r}_{\text{even}} - w_{\text{odd}}\tilde{r}_{\text{odd}}\right),$$
(60)

which can be solved to find $r_0$, $r_{\text{even}}$ and $r_{\text{odd}}$. This analysis has been carried out without periodic modulation of the input. Including periodic modulation with frequency $f$ once again just leads to two equivalent sets of algebraic equations for $v_+ = v + 2\pi f$ and $v_- = v - 2\pi f$. Then, the self-consistent equations for the growth rates of the connectivity are just (*Equations 43–45*), where the $r$s are replaced by the $\tilde{r}$s calculated above.

## Linear stability of spontaneous activity

To study the stability of the spontaneous state we consider (*Equations 51 and 52*) with constant input $I = I_0$. We consider perturbations of the steady state solution of the form $r(t) = r_0 + (\delta r_0 + \delta r_{\text{even}}\cos(\theta) + \delta r_{\text{odd}}\sin(\theta))e^{\lambda t}$, and $x(t) = x_0 + (\delta x_0 + \delta x_{\text{even}}\cos(\theta) + \delta x_{\text{odd}}\sin(\theta))e^{\lambda t}$. Plugging these formulas into (*Equations 51 and 52*) yields a fourth-order characteristic equation for the eigenvalues associated with the spatially modulated modes. In the limit $U_0 \ll 1$ and $1/\tau_x \ll 1$, which is always the case in our simulations, this equation reduces to a quadratic equation, the solution of which is

$$\tau\lambda = -\left(1 - \frac{w_{\text{even}}x_0}{2}\phi'\right) \pm i\frac{w_{\text{odd}}x_0}{2}\phi',$$
(61)

which is similar to the case without synaptic depression, (*Equation 50*), with the sole difference that the connectivity is multiplied by the the depression variable in the steady state $x_0$. Therefore there again is an instability to traveling waves, but the effective recurrent connectivity is weakened by the synaptic depression, thereby shifting the instability. When there is no depression $x_0 = 1$ and we recover the previous result.

## Analysis of data from hippocampal recordings

We analyzed data collected from hippocampal recordings using bilateral silicon probes from two distinct sets of experiments. In the first set two rats had their medial septa reversibly inactivated through injection of muscimol and were then exposed to a novel square, periodic track for 35 min (*Wang et al., 2015*). After a recovery period they were re-introduced onto the same track for an additional 35 min. In the second set of experiments two rats were exposed to a novel hexagonal track for 35 min in a first session. Thereafter they were placed back onto the same track for two additional 35 min sessions with a rest period in between each session.

### Medial septum inactivation experiments

There were a total of n = 124 and n = 128 cells from rats A991 and A992 respectively. In order to identify place cells we first calculated a rate map on the track for each neuron, and then linearized the rate to obtain a one-dimensional place field. We then fit the place field with the function $r = r_0 + r_1\cos(\theta - \phi_1) + r_2\cos 2(\theta - \phi_2)$ thereby extracting the coefficients of the Fourier modes and their phases, see *Figure 7—figure supplement 1a and b* and *Figure 7—figure supplement 2*. We also extracted the coefficients via Fast-Fourier Transform and found identical results. We did not find any cells for which $r_2$ was significant compared to the first two coefficients and hence limited our analysis to the first two coefficients and the phase, that is the place field was centered at $\phi_1$. We excluded all cells for which the tuning index (TI) $r_1/r_0$ was below one. We additionally required that the mean firing rate of selected cells over the entire experiment was greater than 0.4 Hz. With these criteria we had a total of n = 13 and n = 19 cells for the muscimol and post-muscimol sessions in rat

A991 and n = 4 and n = 0 cells for rat A992, see *Figure 7—figure supplement 4*. To ensure we had a sufficient number of neurons we additionally compared the time-resolved SC to the SC from 5000 shuffled orderings of the cells (t-test, p<0.005, corrected for multiple comparison). Significant differences indicate that the SC carries spatial specificity as opposed to simply reflecting the degree of global correlation in the network (shaded regions in *Figure 7—figure supplement 4*). The SC for rat A991 was always significantly different from shuffled for all TI, while for rat A992 it was often not so. For this reason we only used the data from rat A991 for these experiments, see *Figure 7*. We furthermore computed the power spectrum of a simultaneously recorded local field potential (LFP) signal, see *Figure 7c and d* and *Figure 7—figure supplement 4*.

## Hexagonal track experiments

There were a total of n = 158 and n = 134 cells from rats A991 and A992 respectively. We selected cells as described above. This left a total of n = 6, 9 and 4 cells and n = 17, 20 and 13 cells for rat A991 and A992 respectively for the three sessions. The SC calculated for rat A991 was not significantly different from that from 5000 shuffled data sets with random re-orderings of the cells, see non-shaded regions in *Figure 7—figure supplement 3*. Therefore we only considered data from rat A992 for these experiments, see *Figure 7e*.

## Calculating the SC

We consider the firing rate of a $N$ neurons in a time interval of length $T$, divided into $k$ equal time bins of length $T/k$. Once we have ordered the cells according to their place field positions, the cross-correlation of two neighboring cells over the interval is given by

$$CC_{i,i+1}^{\mu} = \frac{1}{T}\sum_{t=1}^{k} \frac{\left(r_i^t - \langle r_i \rangle\right)\left(r_{i+1}^t - \langle r_{i+1} \rangle\right)}{\sqrt{Var(r_i)Var(r_{i+1})}}, \tag{62}$$

where $r_i^t$ is the firing rate of cell $i$ in time bin $t$, and $r_i$ and $Var(r_i)$ are the mean and the variance of the firing rate over the interval. The superscript $\mu$ indicates the particular ordering. The sequential correlation is the average cross-correlation of neighbors in the network

$$SC^{\mu} = \frac{1}{N-1}\sum_{i=1}^{N} CC_{i,i+1}^{\mu}. \tag{63}$$

Therefore we can also write

$$SC^{\mu} = \frac{1}{T}\sum_{t=1}^{k} SC_t^{\mu}, \tag{64}$$

where

$$SC_t^{\mu} = \frac{1}{N-1}\sum_{i}^{N} \frac{\left(r_i^t - \langle r_i \rangle\right)\left(r_{i+1}^t - \langle r_{i+1} \rangle\right)}{\sqrt{Var(r_i)Var(r_{i+1})}}. \tag{65}$$

We can use (*Equation 65*) to calculate the variance of the SC over the time interval in question. The SC in *Figure 7c,d,e* as well as in *Figure 7—figure supplements 3,4*, is calculated by first dividing the total experiment into intervals of 3 min. For each of these 3 min intervals a bin of 1 s was used to calculate the SC. In *Figure 7b* the firing rates over the whole experiment are used, in time bins of 500 ms. The shuffled SC shown in *Figure 7a* (histogram), and c-e (open symbols) is found in the same way by randomly re-ordering the cells.

## Acknowledgments

AR acknowledges helpful discussions with Pablo Jercog.

## Additional information

### Funding

| Funder | Grant reference number | Author |
|---|---|---|
| Ministerio de Economía y Competitividad | BFU-2012-33413 | Alex Roxin |
| Ministerio de Economía y Competitividad | MTM-2015-71509 | Alex Roxin |
| Generalitat de Catalunya | CERCA program | Alex Roxin |
| Howard Hughes Medical Institute | | Yingxue Wang |
| Max-Planck-Gesellschaft | | Yingxue Wang |

The funders had no role in study design, data collection and interpretation, or the decision to submit the work for publication.

### Author contributions
Panagiota Theodoni, Software, Investigation, Writing—review and editing; Bernat Rovira, Software, Data analysis; Yingxue Wang, Resources, Data curation, Writing—review and editing; Alex Roxin, Conceptualization, Software, Formal analysis, Supervision, Funding acquisition, Investigation, Methodology, Writing—original draft, Writing—review and editing

### Author ORCIDs
Alex Roxin [iD] http://orcid.org/0000-0003-1015-8138

### Decision letter and Author response
Decision letter https://doi.org/10.7554/eLife.37388.032
Author response https://doi.org/10.7554/eLife.37388.033

## Additional files
### Supplementary files
• Transparent reporting form
DOI: https://doi.org/10.7554/eLife.37388.021

### Data availability
All electrophysiological data has been uploaded to the Dryad website. The DOI is doi:10.5061/dryad.n9c1rb0

The following dataset was generated:

| Author(s) | Year | Dataset title | Dataset URL | Database and Identifier |
|---|---|---|---|---|
| Theodoni P, Rovira B, Wang Y, Roxin A | 2018 | Data from: Theta-modulation drives the emergence of connectivity patterns underlying replay in a network model of place cells | https://doi.org/10.5061/dryad.n9c1rb0 | Dryad Digital Repository, 10.5061/dryad.n9c1rb0 |

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
