## [Decision Letter]

Thank you for submitting your article "Theta-modulation drives the emergence of connectivity patterns underlying replay in a network model of place cells" for consideration by *eLife*. Your article has been reviewed by three peer reviewers, one of whom is a member of our Board of Reviewing Editors, and the evaluation has been overseen by Michael Frank as the Senior Editor. The reviewers have opted to remain anonymous.

The reviewers have discussed the reviews with one another and the Reviewing Editor has drafted this decision to help you prepare a revised submission.

Summary:

Overall the reviewers thought that this was an interesting study. However, a substantial number of issues were identified, and in subsequent discussions all reviewers thought each other's revision requests were valid and important.

We believe that it should be manageable to address them within a reasonable time.

Essential revisions:

Here is the summary of the essential revisions. These are just summaries, for the full comments see below.

- Discuss, and ideally simulate, the effect of phase precession.

- More clearly discuss the requirement that STDP rule cannot be frequency dependent, as well as the exact prediction regarding the optimal frequency.

- Extend the discussion of the result of Equation 5 and its predictions.

- Discuss, and ideally simulate, reverse replay in the model.

- Clarify the brain region which is modeled from the outset.

Reviewer #1:

In this study the authors present a rate based network model representing the place field activity in rodents on a periodic track. An external input activates place cells according to their preferred position and an asymmetric learning rule strengthens the recurrent excitatory synaptic weights such that during bursts in a quiet period (no place cell input) replay of the place cell sequence occurs. This learning is accelerated by theta oscillations. Over all, this is a well done study, consisting of computer simulations, analytical derivations and the analysis of electrophysiological data. However there are a few comments that should be addressed.

The model assumes a Poisson-rate for each neuron. However, it has been shown that place cells phase precess during theta oscillations which leads to a sequential organization of spiking during the theta cycle. This phenomenon is not mentioned at all. What effect does phase precession have on the presented results?

In this model the growth rate of the symmetric mode, which governs the learning process, depends on the auto-correlation and the plasticity rule. But the auto-correlation itself depends on the firing rate of the place cell. What happens with heterogeneous peak rates as shown in Figure 7A? Does this framework still hold?

I wonder whether the muscimol experiments are sufficient to claim the importance of theta oscillations for the learning process. A counter-example for place cell activity in the absence of theta oscillations has been demonstrated in bats (Yartsev and Ulanovsky, 2013), which should be mentioned. Yes, silencing the medial septum largely abolishes theta oscillations, however, it might be a stretch to reduce the medial septum to its pace making function.

Reviewer #2:

Summary:

The authors describe a spiking neural network model of place cells using what seems to be a variation of Linear-Nonlinear-Poisson (LNP) neurons equipped with a temporally asymmetric Spike-Timing-Dependent Plasticity (STDP) rule and externally oscillatory modulation. The network is capable of giving rise to spontaneous replay following simulated exploration, and its learning rate is optimized by external modulation in the Theta range. Moreover, the authors performed an analytical treatment of a simplified version of the network model to derive a formal description of how the robustness of and timescale in which replay emerging from repeated pre-post spike pairings is modulated by oscillatory activity at various frequencies.

The science behind the paper is very solid, and the analytical treatment in addition to the numerical simulations make this a very thorough, impressive modeling paper. The major criticisms revolve around how the authors chose to present some of their findings. Certain claims and findings are very exaggerated or misleading (see Major Critiques 1-3), and it is unclear what brain region are actually modeled (see Major Critique 4). Finally, there is an unsubstantiated claim about the model and oversight of extremely similar previously published work in the Discussion.

Major Critiques:

1) The authors consistently refer to the frequency band which optimizes the learning rate in the model as "theta", which is typically referred to as a hippocampal oscillation within the 8-12 Hz range. However, Figure 4A appears to show that optimal synaptic weight changes occur over a much broader frequency range, and Figure 4B specifically shows that optimal replay occurs within the 2-8 Hz range. The authors should be clear in the introduction about what is typically considered the theta range, and that, while their model does demonstrate the utility of oscillations within the theta range, the effect they observe is not limited to theta.

2) The authors should spend some time interpreting the analytical result from Equation 5 which demonstrates that "the growth rate of the even mode is found by multiplying the AC of place-cell activity by the window for plasticity, and integrating." No real physiological/functional interpretations/implications of this result are mentioned (which should always be done), and this leaves it unclear to readers (particularly ones without a rigorous computational background) how impactful this result actually is. It is unsurprising that integrating the autocorrelation of place-cell activity multiplied by the plasticity window provides the expected growth rate. This is essentially saying that by considering all of the place-cell activity which occurs regularly, and the corresponding weight change given by STDP, while ignoring the noisy activity (which would tend to cancel out on average from this type of unsupervised learning rule) it is possible to estimate the growth rate of the even mode (i.e. the one relevant to learning and which governs replay). The straightforward intuition of this result should be discussed both so that less computationally inclined readers can appreciate the result, and so that it does not mistakenly get taken to be predictive of a novel interpretation of how STDP leads to learning.

3) The authors propose an unsubstantiated mechanism for how their network could also generate reverse replay in addition to the more commonly studied forward replay. They should either provide supporting data for this claim or go into a much more detailed explanation of their proposed mechanism. To begin with, it is unclear what exactly is meant by "synapses to downstream neurons become rapidly depressed". Is this referring to long-term depression, short-term presynaptic depression, etc.? Assuming short-term presynaptic depression (as it has the appropriate timescale) this mechanism would only make sense if the same place-cell sequences were activated during the theta cycle as those immediately reactivated in during sharp-wave replay. However, as far as I know, no such tight temporal correlation has been observed.

4) The authors need to be clearer from the beginning which region of the brain is modeled. The paper does not spend any time on this or the general architecture of the network model in the Introduction. The constant reference to "place-cells" implies that this is a model of CA1, however the connectivity of the model makes this more likely to reflect CA3; a fact which the authors bring up themselves in the Discussion. Their defence of how these results can still apply to learning and replaying place-cell sequences is solid, and I would recommend the authors take this approach from the start. Bringing it up at the end of the paper seems somewhat misleading and could potentially turn off readers and make them dismissive of the interesting findings reported in the paper.

Reviewer #3:

This paper presents a theoretical study how STDP together with short-term synaptic depression and correctly tuned oscillation leads to replay like activity of hippocampal place cells. I think that none of the particular mechanisms are new, but the combination and its application to hippocampal sequences is and the analysis of the complete system is nice.

- Were any special settings needed to make sure the rotation of the attractor stopped (Figure 2C bottom), and did not continue rotating? Or would sufficient experience lead to longer and longer sequences?

- The maximum in Figure 4A hinges on the lack of frequency dependence in the STDP rule. The problem is that, AFAIK, STDP always has shown frequency dependence (more LTP at higher frequencies) and it is hard to imagine how biophysically a precise balanced STDP model could be constructed.

Currently it is written as if this is a minor issue, and that STDP needs some stabilization anyway. That might be true but the maximum might be lost. I think this weakness of the model needs to be exposed more.

Secondly, related but a bit less important, I would expect that even with balanced STDP the weights diverge due to the correlation contribution to LTP.

---

## [Author Response]

Essential revisions:Here is the summary of the essential revisions. These are just summaries, for the full comments see below.- Discuss, and ideally simulate, the effect of phase precession.

Phase Precession

The presence of phase precession in the model

It turns out there is, in fact, some degree of phase precession in our model. Because the model is stochastic, the phase precession should be understood as a precession in the probability of generating a spike (the peak firing rate) with respect to the underlying theta phase as the virtual animal runs through a given place field. This mechanism is explained in detail in (Romani and Tsodyks, 2015) Figure 5. Our model is essentially the same, with the addition of STDP, so the mechanism also holds, with some caveats. This mechanism is intimately tied to the generation of theta sequences during movement, which occurs once the connectivity becomes sufficiently modulated in a novel environment. That is, there is initially no phase precession in the model, just as there is no “replay”. It emerges as the connectivity is shaped through plasticity. We give a detailed description of this effect below.

The mechanism generating phase precession in our model is a network mechanism which was first proposed by Misha Tsodyks and others involving the generation of propagating activity during the theta cycle; this propagation was due to an assumed asymmetry in the recurrent connections between neurons which represented place cells on a linear track (Tsodyks et al., 1996). A very similar effect can be achieved through short-term synaptic depression even without the need for asymmetric connectivity. The reason is that the motion of the animal, and hence the sequential activation of place cells, generates an asymmetric pattern of synaptic depression (upstream synapses are more depressed than downstream ones). This can also lead to theta sequences.

The relationship between theta sequences and phase precession can be illustrated in a cartoon, in which it also becomes very clear that such a mechanism relies on asymmetry in the place cell activity. Author response image 1 shows the collective activity from a population of place cells ordered according to their place field locations (the ovals can be thought of as contours of a firing rate map). The bump of activity reflects the animal’s motion and is modulated by an ongoing theta rhythm. We consider the firing rate of a single place cell, with a place field centered at the position shown by the horizontal dashed blue line. In this example there is no temporal asymmetry in the place cell activity and hence the maximal firing rate of all cells coincides with the peak in the theta rhythm, including that of the blue cell. The peak in the firing rate gives the most likely time for generating a spike, shown by the solid vertical lines. There is no phase precession.

Author response image 1 illustrates the effect of asymmetry in the place-cell activity. Now, within each theta cycle there is a sequence of activation of the place cells. Because of this the maximum firing rate of the blue cell within each cycle does not necessarily coincide with the peak in the theta rhythm. In fact it moves to earlier phases as the animal moves out of the field. This generates phase precession. This is the same mechanism as shown in Figure 5 of (Romani and Tsodyks, 2015). Clearly the details of the phase precession depend on the exact shape of the place-cell activity.

**Author response image 1. respfig1:** Temporally asymmetric place-cell activity leads to phase precession. (**A**) When place-cell activity is symmetric, and hence there are no theta sequences, there is also no phase precession. (**B**) The presence of theta sequences generates phase precession.

This mechanism is present in the model we study. This is shown in Figure 2—figure supplement 5. Specifically, during early exploration the recurrent connectivity is unstructured w.r.t. the space of the novel track. The place cell activity is therefore dominated by sensory input; activity is symmetric, Figure 2—figure supplement 5A, there are no theta sequences, Figure 2—figure supplement 5B, and there is no phase precession, Figure 2—figure supplement 5C. Note also that the place cell activity is highly heterogeneous due to the random nature of the recurrent connectivity from previous learning. So in Figure 2—figure supplement 5 it is even often difficult to define a preferred phase for firing. On the other hand, after sufficient plasticity, the recurrent connectivity becomes strongly modulated in the novel space. When this occurs, short-term synaptic plasticity leads to strong temporal asymmetry in the place-cell activity, Figure 2—figure supplement 5D, there are pronounced theta sequences, Figure 2—figure supplement 5E, and phase precession, Figure 2—figure supplement 5F.

In the case of our model, as in (Romani and Tsodyks, 2015), the phase precession resulting from the synaptic-depression-driven theta sequences, is not as pronounced as that seen in experiment. Therefore, theta sequences are probably not the whole story. In fact, the relationship between theta sequences and phase precession has been known for over a decade (Foster and Wilson, 2007). In that work, the authors show that theta sequences cannot be predicted solely based on the observed phase precession, which may imply the opposite as well. So additional mechanisms are likely.

The role of phase precession in plasticity

The weak phase precession already present in the model arises during the plasticity process, as the recurrent connectivity becomes significantly modulated. Therefore it is a result of the plasticity process. In order to study the effect of phase precession on the learning process itself, we can introduce it by hand by adding an additional term to the external input to place cells. This allows us to study the role of phase precession parametrically, assuming it is an extrinsic effect. The external forcing now takes the form(1)Iext=I0+IPFcos(θ−vt)(1+Ithetacos(2πft−αθ))

The new term, proportional to *α*, generates theta sequences. Note that when *α* = 0 we recover the previous forcing, for which the maximum of the input is simply *θ_max_*= *vt*, i.e. it follows the motion of the animal. On the other hand, when *α* ≠ 0 the maximal input sweeps ahead of the position of the animal periodically. For example, for *α* = 1 it can be shown that the maximal input is *θ_max_*= *vt* + *ψ* where *ψ* = *πft* mod 2*π*. Author response image 2 shows place cell activity and input for two values of *α* = 0,1 which correspond to theta activity without and with theta sequences respectively.

**Author response image 2. respfig2:** Place cell activity and place cell input with and without theta sequences. (**a**) Place cell activity with no theta sequences (*α* = 0). (**b**) Input with no theta sequences. (**c**)-(**d**) Same as (**a**) and (**b**), but with theta sequences (*α* = 1). Color indicates activity or input in Hz.

As it turns out, generating theta sequences, which is the rate-based equivalent of phase precession, has a significant effect on the growth rate of the even mode of the connectivity. It therefore also strongly influences the time at which a transition to replay occurs. Specifically, theta sequences speed up learning. This is shown in Author response image 3 which shows that the transition to replay occurs earlier when *α* = 1 compared with *α* = 0. The inset quantifies the dependence of the transition time on *α*. The curve asymptotes for large enough *α*.

Why is there a speed-up in learning when theta sequences are present? We can solve Equations 39 and 40 of the Materials and methods using Equation 1 as the external input to see how the growth rate of the even modes of the connectivity are affected by *α*. Analytically this is quite messy. The reason is that for *α* = 0, which is the case studied in the paper, the input only has the spatial Fourier modes cos*θ* and sin*θ*, and a self-consistent solution to Equations 39 and 40 can be found by considering only these. When *α* ≠ 0, additional spatial modes appear, specifically cos((1 ± *α)θ*), sin((1 ± *α)θ*). This means that a self-consistent solution will involve at least these modes, making the calculation extremely lengthy and tedious.

**Author response image 3. respfig3:** The SC as a function of time for two simulations: *α* = 0, i.e. no theta sequences and *α* = 1, i.e. with theta sequences. Learning is faster and hence the transition occurs earlier with theta sequences. The transition time decreases monotonically as a function of *α*, which sets the velocity of the theta sequences.

Therefore, rather than do this we performed numerical simulations of the full model, using Equation 1 as an input, but with a linear-threshold transfer function. We also take *I*_0_
*> I_PF_*which ensures that the firing rate dynamics operates in the linear regime. Doing this makes the full system equivalent to Equations 39 and 40, which assume a linear transfer function. When we do this we find that taking *α* ≠ 0 indeed leads to the growth of additional spatial modes in the connectivity (not shown). However, the amplitude of the even modes is actually smaller than in the *α* = 0 case. On the other hand, when we take *I*_0_
*< I_PF_*, so that the firing rate dynamics operates in the nonlinear, rectified regime, the opposite occurs. Namely, when *α* ≠ 0, the even modes grow faster than when *α* = 0. This strongly suggests that the speed-up of learning due to the theta sequences is a nonlinear effect, not captured by our current theory.

Summary of phase precession

We apologize for the length of our response, but it turned out that phase precession/theta sequence dynamics in our model are quite rich. In short, we can make the following two statements: 1 – theta sequences emerge spontaneously in our model during the learning process, leading to weak phase precession. 2 The presence of theta sequences during learning speeds up the learning process. The mechanism is nonlinear but not well understood (at least by us) at this point. Also note that the role of the “theta” frequency *f* on the growth rate of the even mode, continues to hold even in the presence of theta sequences. So theta sequences represent a means of providing an additional speed-up of learning once the theta rhythm is already there, in our model.

Changes to Manuscript

We have now added two sentences to the Results. They are: “This steady increase is due to the emergence of so-called theta sequences. […] The emergence of theta sequences here also coincides with a precession of the phase of the underlying firing rate with the ongoing theta rhythm, see Figure 2—figure supplement 5f and Discussion for more detail.”

We have also added a new section in the Discussion on theta sequences and phase precession.

- More clearly discuss the requirement that STDP rule cannot be frequency dependent, as well as the exact prediction regarding the optimal frequency.

Frequency dependence of the STDP rule

Balanced rule

In this paper we have only considered a balanced STDP rule, i.e. one for which the strength of potentiation and depression are on average the same given fixed pre- and post-synaptic firing rates. The upshot for our analysis is that only time-varying firing rates lead to lasting network-wide changes in the synaptic connectivity. Specifically, if we write the recurrent connectivity profile as(2)w(Δθ)=w0+wevencos⁡(Δθ)+woddsin⁡(Δθ)then with a balanced plasticity rule *w*_0_ will not change. When this is the case then the growth of *w_even_*depends crucially on the presence of an ongoing oscillation in the place-cell activity, as explained in the main text of the paper. It can be shown that with a pairwise STDP rule the maximum growth rate occurs for a frequency *f* = 1*/*(2*π*√*τ*_+_*τ*−). This can be found by taking the derivative of Equation 23 with respect to the velocity term and plugging in *v* = 2*πf*. From this formula it is easy to see that temporal windows for the STDP rule on the order of 10s to 100s of ms will lead to best frequencies between 1 and 10 Hz. So over the physiological range of observed STDP windows the best frequency is within the range of theta. As noted by the reviewers, the growth curve does not drop off precipitously away from the best frequency. For example, in Figure 5C of the manuscript, frequencies in the range 3-12Hz more or less give similar growth rates. Therefore, depending on the STDP time constants there will be some range of frequencies for which the growth rate is high. For much lower and much higher frequencies the growth rate does fall by orders of magnitude. For example, again in Figure 5C, a frequency of 1Hz or 30Hz gives an order of magnitude lower growth rate. So if an 8Hz modulation leads to a transition to replay after 20min, then a 1Hz modulation would take 3hrs and 20 min. This is a large difference. If a balanced triplet rule is used the best frequency could be found by taking the derivative of Equation 26 w.r.t. the velocity. We cannot find an analytical solution, but Figure 5E of the main text suggests the best frequency is close to the one from the pairwise rule.

Unbalanced rule

If the rule is not balanced then *w*_0_ will grow or decay if potentiation or depression dominates respectively. In this case the bounds on the synaptic weights become very important. In particular, if we bound the weights below by a minimum and above by some maximum weight, then with the unbalanced rule, after sufficient time all of the synapses will be fully potentiated or fully depressed. Therefore no structure is possible in this network with an unbalanced rule. This is a very general finding in recurrent networks. In fact, even to store simple patterns using multistable fixed points in networks with STDP is quite hard, and requires additional fast homeostatic mechanisms, e.g. (Zenke et al., 2015).

On the other hand, in-vitro experiments have shown a “frequency-dependence” of the stimulation protocol on the shape of the STDP window. We would like to emphasize that the frequency in question here is the repetition-rate, not the frequency *f*, i.e. the modulation or “theta” frequency from our model. Rather, it would correspond to the pre- and post-synaptic firing rates. So from now on we will talk about a rate-dependence of the STDP rule just to be clear. This rate-dependence can be taken into account using a triplet STDP rule (Pfister and Gerstner, 2006). This rule is unbalanced in the sense that at high rates potentiation dominates and so using the rule as is will lead to a fully potentiated network. We can see from our Equation 26 that taking a sufficiently unbalanced rule can eliminate much of the dependence of the growth rate of the even mode on the modulation frequency. Specifically, if we take A3+ much larger than the pairwise parameters *A*_+_ and *A*− and A3- = 0, then the growth rate will be large and positive even if there is no modulation whatsoever. Unfortunately, if we do this then Equation 25 shows that *w*_0_ will also grow very fast, wiping out all structure in the network.

We can take a less extreme example to make the point. Author response image 4 of this reply shows the dependence of the growth rate of the homogeneous mode, i.e. the mean synaptic weight, as a function of the mean firing rate of the place cells. Note that this firing rate dependence is what would be called “frequency dependence” of the STDP rule; here we all it rate dependence. In order to calculate this curve, we use Equations 25-27 of Materials and methods but including all of the terms which depend on the modulation frequency *f* as described in the section “including periodic modulation”. The parameters used are *τ*_+_ = 20ms, *τ*− = 60ms, *A*_+_ = 0.001, *A*− = 0.001*/*3, *v* = 0.001, *τ* = 20ms, *τ_x_*= *τ_y_*= 100ms, *A*+3 = 10−7, *A*3− = 10−8*/*3.

Again, for the balanced rule (black line in Author response image 4), there is no rate-dependence of the growth rate of the mean connectivity. This is what is meant by balanced. In the unbalanced case (orange line) there is a dependence on rate. Author response image 5 shows the dependence of the even mode and the mean connectivity on the modulation frequency if we fix the mean firing rate at *r*_0_ = 10Hz. Both the balanced and unbalanced cases show a clear maximum in the growth rate of the even mode. On the other hand, for the unbalanced case the growth rate is significant for any frequency whereas in the balanced case it goes to zero at low and high frequencies. Importantly, the growth rate of the mean synaptic weight *w*_0_ is positive at any frequency. So again in the unbalanced case, after sufficient time all of the synapses will become maximally potentiated, and no network structure is possible. This example merely serves to illustrate the fact that we cannot study an unbalanced rule in a recurrent network without including additional homeostatic mechanisms.

**Author response image 4. respfig4:** Growth rate of the mean connectivity as a function of the average neuronal firing rate.

**Author response image 5. respfig5:** Growth rate of the even mode (top) and mean connectivity (bottom) as a function of modulation frequency *f* with a balanced rule (black) and an unbalanced rule (orange). *r*_0_ = 10Hz.

Unbalanced rule with additional mechanisms

Therefore, in order for non-trivial structure to evolve in a recurrent network with unbalanced STDP, additional mechanisms must be included. In (Zenke et al., 2015) this includes a depressing heterosynaptic mechanism, Equation 13 in that paper. Studying that particular mechanism in the context of our model is possible, but would represent a major undertaking. The reason is that heterosynaptic term is proportional to the difference between the current value of the synaptic weight and a reference value which itself changes in time. Furthermore, the pairwise depression term is subject to slow homeostasis, Equations 17-18 in that paper. So we unfortunately cannot give a detailed answer here regarding the robustness of our result for that rule. On the other hand, in order to allow for the development of non-trivial structure there should be some mechanism, on not too long a time scale, which brings down the orange curve in Author response image 5 (bottom). If this mechanism is nonlinear in the firing rates, which is the case for (Zenke et al., 2015) for example, then it will also cause the orange curve in Author response image 5 (top) to shift downward. In the limit of very fast “homeostasis” this will just be equivalent to the black curves. For non-instantaneous mechanisms (the relevant case) a systematic study of the dynamics would be needed.

Changes to manuscript

We have now added three sentences to the Discussion, under “Robustness to changes in the plasticity model and to the presence of spike correlations”. They are: “Specifically, when the repetition-rate of spike pairs is high, in-vitro STDP is dominantly potentiating; for such a rule there would be no non-monotonic dependence of the learning rate on modulation frequency at high rates. […] It is unclear if the learning rate of this rule would also exhibit a non-monotonic dependence of the modulation frequency, as in our rule.”

- Extend the discussion of the result of Equation 5 and its predictions.

Discussion of Equation 5

Equation 5 is one of the main results of the paper, so we appreciate the chance to discuss it more in depth. It is easiest to first discuss the plasticity between pairs of neurons in this model. This is given by Equation 3 of the manuscript. If we average this equation to eliminate fluctuations on a fast time scale we have(3)w˙(θ,θ′)t=∫−∞∞⁡dTK(T)⋅CCθ,θ′(T)where *CC _θ,θ’_ (T) = ⟨r (θ’, t) r (θ, t + T)⟩ _t_*. Author response image 6 shows how the integral can be evaluated for the hypothetical example in which the cell at *θ*′reliably fires before the cell at *θ*. In this case the likelihood of a spike pair peaks at 25ms. Author response image 6 show that the synapse from *θ*′to *θ* will potentiate while that from *θ* to *θ*′will depress. Although Equation 3 is useful for determining the evolution of the synaptic weights between a single pair of neurons, this framework is not very helpful in a large recurrent network. In a network of *N* place cells there are 2*N* such equations for the synaptic weights, not to mention the corresponding equations for the firing rates.

The great simplification comes about by decomposing the recurrent connectivity into its Fourier components, specifically

w(θ,θ)=w(θ−θ)=w0+wevencos(θ−θ)+woddsin(θ−θ)

Here the coefficients *w*_0_, *w*_even_ and *w*_odd_ are not the synaptic weights between pairs of neurons. Rather, *w*_0_ is the mean synaptic weight in the network, *w*_even_

is the amplitude of a pattern of synaptic weights in which nearby neurons are strongly coupled in both directions, and more distant neurons have effectively inhibitory interactions. It is often what is called a “Mexican hat” connectivity profile. *w*_odd_ is a pattern in which nearby neurons are strongly coupled in one direction and effectively inhibitory in the other. We can isolate the Mexican

hat component of the connectivity by taking *θ*′= *θ*, i.e. by considering place cells with overlapping place fields and subtracting off the mean connectivity, *w*_0_. This is because only growth of the even mode leads to strong recurrent excitation locally. If we now take the temporal derivative of the connectivity and average in time we arrive at Equation 5, which therefore gives the rate of change of the amplitude of the even mode, or the Mexican hat-like connectivity profile. The derivative of *w*_0_ vanishes because it is a balanced STDP rule.

**Author response image 6. respfig6:** Changes in individual synapses depend on the cross-correlation of the firing rate of pre- and post-synaptic cells. (**a**) The STDP window *K(**T***) (red and blue curves) and the cross-correlation *CC_θ,θ_*^′^(***T***) (dashed black line) for two hypothetical place cells. (**b**) The product *K(**T***) · *CC_θ,θ_*^′^(***T***). (**c**)-(**d**) Same as (**a**)-(**b**) but for *C_θ_*^′^*_,θ_(**T**).* The rate of change of the synaptic weight is found by integrating the curves in (**b**) and (**d**) respectively.

Biological interpretation of Equation 5

The biological interpretation of Equation 3 is straightforward. If you want to know how the synapses change between a pair of cells, you should just count all spike pairs and use the STDP rule to look up all potentiations and depressions. If the neurons’ activity is a noisy but stationary process, then this prescription is equivalent to first using the full spike trains to calculate the cross-correlations (CC). Then you just multiply the CC by the STDP window and integrate. Again, this is the same as adding up all of the changes due to each and every spike pair. But it is much easier to implement with real data (assuming you know what the STDP rule is).

The biological interpretation of Equation 5 (from the paper) is less straightforward, but is the following. If you are doing an experiment in which you believe the connectivity between any two place cells is most strongly dependent on the distance between their place fields alone, and not, say the absolute position of any place field, then you can decompose the connectivity into a series which depends only on this distance (in our case we have considered data from a periodic track which the animal explores more or less homogeneously, so there are no “special” locations. If there is a special location, such as a rewarded location, then this theory can be extended by separating the connectivity into a part which only depends on differences between locations, and a location-specific (e.g. reward specific) part. But that goes beyond the scope of this paper). If this is the case, then you can actually infer global properties of the network structure just by looking at the neuronal activity locally. Specifically, if you find that the activity of neurons with overlapping (or very nearby) place fields would induce strong reciprocal potentiation, then you can conclude that a Mexican-hat like connectivity is forming. Theoretically we assume a large number of place cells with arbitrarily close place fields. In the limit of many cells, the CC of very nearby cells just becomes the auto-correlation AC. Therefore, one can simply take the full spike train of each place cell, calculate the AC, and multiply it by the STDP window. Given real data, for which the AC of each cell will be different, one could take the ensemble average to get the mean growth rate of the even mode. In our theoretical framework each cell has an identical AC by construction.

Now consider the case where the STDP window is perfectly anti-symmetric, with identical depression and potentiation lobes. The AC is by definition symmetric. Therefore, the product of these functions is also anti-symmetric (or odd) and so the integral is zero, i.e. there is zero growth of the even mode of the connectivity. What is the biological interpretation of this? If we consider place cells with very nearby place fields, then as the animal moves through the place fields in one direction the synapse from cell A to cell B will be strengthened on average, while those from B to A will be weakened. We are taking about changes due to spike pairs with latencies which are most likely to be relatively small, since the place fields are overlapping. If the animal only ever moves from A to B, then the synapse from A to B will continue to get stronger and from B to A weaker. If the animal moves equally in both directions then there will be *no* overall change in synaptic strength. Therefore, there will be no strong reciprocal excitatory connectivity in the network.

Changes made to manuscript

We have now added a detailed description of the biological interpretation of Equation 5, directly following Equation 5 in the main text. It reads: “Note that despite the similarity in form between Equation 5 and Equation 3, the biological interpretation of the two is quite distinct. […] The key assumption that makes this possible is that the synaptic weight between any two cells should depend only on the difference in place-field location and not on the absolute position, Equation 4.”

- Discuss, and ideally simulate, reverse replay in the model.

Reverse Replay

Replay in the model occurs spontaneously once the recurrent connectivity is sufficiently modulated through plasticity in any given environment. Specifically, once the even mode of the connectivity reaches a critical value, then traveling waves emerge spontaneously through an instability of the homogeneous state. The mathematics behind this are described in the section “A traveling wave instability of the rate equation for modulated connectivity” and “Including the effect of synaptic depression” of Materials and methods. Equations 50 and 61 show that the velocity of the traveling wave is proportional to the odd mode *w*_odd_. Therefore, if, during the exploration of a novel environment, the recurrent connectivity is asymmetric, meaning the odd mode is not zero, then waves will appear which travel only in one direction. How then can one get replay in both directions?

Reverse replay through local input

The solution we propose in the discussion is to allow for strong local input. Such input can lead to activity propagating backward w.r.t. the asymmetry in the connectivity. Strong, local input, presumably at or near the animal’s location, leads to depletion of synaptic resources in the recurrent connections. Because the connectivity is asymmetric, this depletion is also asymmetric, more strongly affecting the direction in which replay would usually propagate. Therefore, the activity is now forced to propagate backward. A simulation illustrating this can be seen in Figure 2—figure supplement 7. One appealing feature of this hypothesis is that it requires spatial input to work. In fact, during sleep, when such sensory input is missing, replay is predominantly forward (Roumis and Frank, 2015). Backward replay is more often observed during awake quiescence when such input *is* available, and may be mediated by place cell activity in CA2 (Kay et al., 2016). So the mechanism is consistent with those particular findings. On the other hand, and as can be seen in Figure 2—figure supplement 7, the backward replay which arises via this mechanism does not look the same as the forward replay. Differences include spiking intensity, duration, and velocity. To our knowledge there has not been any systematic study of these properties comparing forward and backward replay in the experimental literature. Additionally, such differences may not be so striking unless many cells are recorded simultaneously.

Reverse replay with symmetric connectivity

When the connectivity is perfectly symmetric, once the even mode reaches a critical value there is an instability to a stationary spatial bump, and not a wave. This can be seen in Equations 50 and 61 by taking *w*_odd_ = 0. Nonetheless, in the presence of short-term synaptic depression this stationary bump is not stable, and traveling waves occur anyway. These traveling waves emerge from the bump state and not the homogeneous state, and as such are not amenable to analysis. We can study them numerically however. Author response image 7 shows how the spontaneous activity changes when the connectivity is perfectly symmetric. Panels (A)-(C) show the typical activity for our model, after exploration of 10 environments. This is just a reproduction of Figure 2—figure supplement 2 of the original manuscript. Panels (D)-(F) show how the dynamics changes when the connectivity from (A)-(C) is made symmetric by taking the connection from cell *j* to cell *i* to be (*w_ij_*+ *w_ji_)/*2 where the *w*s are from the learning process. In this way *w_ij_*= *w_ji_*. Note that in Author response image 7 the replay is always in the same direction, whereas in Author response image 7 the activity can propagate both ways. This is just an illustration. The statistics of the replay process for the symmetric case can be found in Figure 2 of (Romani and Tsodyks, 2015).

When would the connectivity be perfectly symmetric? In our model the connectivity would only be symmetric if the animal explored equal times in the clockwise (CW) and counter-clockwise (CCW) directions. This seems unlikely. Indeed, in the data we have there is a clear bias in the animal’s movement for CW motion. Alternatively, there could be some homeostatic process which would conspire to make the connectivity symmetric.

Summary

Plasticity generates asymmetric recurrent connectivity in general in our model. On a periodic track there is no global remapping of place cells when the animal switches direction. Therefore, equal exploration of CW and CCW directions can lead to symmetric connectivity, which would give rise to forward and backward replay. Otherwise, replay goes in one direction, and strong local input is needed to generate replay in the other direction. On a linear track only this second mechanism would work due to global remapping.

Working out the details of replay, including differences between forward and backward replay statistics in data, and also between linear tracks and periodic tracks, would be a valuable contribution. It would also be a huge undertaking. Our proposed mechanism is therefore not a well-fleshed out theory, but merely an idea, albeit one which we have shown to work in our model. Therefore, while we felt it important to mention, we did not deem it conclusive enough to add to the main text. We felt a paragraph in the Discussion was fitting.

Changes to manuscript

We have altered the description of the mechanism for backward replay in the Discussion in order to clarify it. It now reads: “Specifically, if global input to our model network is homogeneous then replay occurs only in one direction. […] This forces the activity to travel backward with respect to the bias in the connectivity.”

The simulations in Figure 2—figure supplement 8 provide an illustration of this effect.

**Author response image 7. respfig7:** The spontaneous dynamics in a model with asymmetric versus symmetric recurrent connectivity. (**a**) The amplitude of spontaneous activity as a function of the external drive (uniform, non-specific drive to all place cells) in an asymmetric network. This is a reproduction of Figure 2—figure supplement 8; the parameters are all the same. (**b**) A space-time plot of the activity. Replay is always in one direction. (**c**) Blow-up of the replay activity. (**d**)-(**f**) The same for a network with symmetric recurrent activity. In this case we take the synaptic connection from a cell *j* to a cell *i* to be ˜*w_ij_*= (*w_ij_*+ *w_ji_)/*2, where *w_ij_*and *w_ji_*are the weights after the exploration of 10 tracks for one hour each. These are exactly the (asymmetric) weights used in (**a**)-(**c**). The external input is *I_ext_*= −0.8.

- Clarify the brain region which is modeled from the outset.

Clarifying the brain region

Our model is definitely a model of a strongly recurrent circuit, such as in CA3 and not in CA1. We now mention this explicitly at the outset of the paper. Specifically, in the Introduction we now state:

“Here, we develop a model that explains how synaptic plasticity shapes the patterns of synaptic connectivity in a strongly recurrent hippocampal circuit, such as CA3, as an animal explores a novel environment.”

At the beginning of the Results section we say:

“We modeled the dynamics of hippocampal place cells in a strongly recurrent circuit, such as area CA3, as an animal sequentially explored a series of distinct novel ring-like tracks”.

Unsolicited changes

On revising the manuscript we realized that the sequential correlation (SC) was not formally defined. We therefore added a brief section to Materials and methods to describe this.

Reviewer #1:In this study the authors present a rate based network model representing the place field activity in rodents on a periodic track. An external input activates place cells according to their preferred position and an asymmetric learning rule strengthens the recurrent excitatory synaptic weights such that during bursts in a quiet period (no place cell input) replay of the place cell sequence occurs. This learning is accelerated by theta oscillations. Over all, this is a well done study, consisting of computer simulations, analytical derivations and the analysis of electrophysiological data. However there are a few comments that should be addressed.The model assumes a Poisson-rate for each neuron. However, it has been shown that place cells phase precess during theta oscillations which leads to a sequential organization of spiking during the theta cycle. This phenomenon is not mentioned at all. What effect does phase precession have on the presented results?

Although our neurons are Poisson we can still talk about the order in which neurons fire in terms of the underlying likelihood of spiking, i.e. the rate. Therefore, we can still study the phenomenon of phase precession, defined in this way. It turns out that: 1) Theta-sequences emerge in our model during learning. A signature of these sequences is phase precession, although it is a weak effect. 2) Introducing phase precession by hand during the learning process, in the form of theta-sequences, significantly speeds up learning. These two findings are described in depth above, under “Essential revisions”, in the section entitled “Phase precession”.

In this model the growth rate of the symmetric mode, which governs the learning process, depends on the auto-correlation and the plasticity rule. But the auto-correlation itself depends on the firing rate of the place cell. What happens with heterogeneous peak rates as shown in Figure 7A? Does this framework still hold?

A very relevant question. In our theory, in order to facilitate the analysis, we assume that place fields are homogeneously distributed and identical in shape. In the data this is clearly not the case. In the model there is some variability in place field shape due to heterogeneity in the recurrent connectivity, but it is largely homogeneous because each place-cell receives place-field input of the same shape.

We have tested the effect of heterogeneity by allowing for a distribution of place-field input to place cells. Specifically, we distribute the amplitude of the place-field input *I_PF_*. Figure 2—figure supplement 4A shows how heterogeneity affects the transition to replay. The curve with the black circles is identical to the simulation in Figure 2 of the main text, for which *I_PF_*= 25*Hz*. The red squares are a simulation for which *I_PF_*is uniformly distributed between 20 and 30Hz (and hence the mean is the same as before). The orange diamonds show an extreme case where *I_PF_*is uniformly distributed between 0 and 50Hz. Figure 2—figure supplement 4B shows sample place cell activity for the strongly heterogeneous case. Note that in this case some cells are only very weakly selective to place, e.g. cell 3, while others have no place field whatsoever, e.g. cell 4. Therefore, heterogeneity in place cell activity has a quantitative effect on the learning dynamics and resulting transition, but does not alter the results qualitatively.

We have now added a sentence to the main text which states:

“This transition [in SC] occurred even in simulations where we included strong heterogeneity in place field tuning, see Figure 2—figure supplement 4.”

I wonder whether the muscimol experiments are sufficient to claim the importance of theta oscillations for the learning process. A counter-example for place cell activity in the absence of theta oscillations has been demonstrated in bats (Yartsev and Ulanovsky, 2013), which should be mentioned. Yes, silencing the medial septum largely abolishes theta oscillations, however, it might be a stretch to reduce the medial septum to its pace making function.

This is a fair point. Inactivating the medial septum undoubtedly has other effects beyond reducing theta. Some of these effects we have characterized to some extent in the supplementary figures. For example, there is stronger rate remapping of place fields when the animal switches from CW to CCW motion under muscimol, see Figure 7—figure supplement 1C. Firing rates are also reduced, see Figure 7—figure supplement 2A. Both of these effects are relatively weak and neither would qualitatively alter any conclusions drawn from our computational model. On the other hand in the experiments we have looked at theta is dramatically reduced, essentially eliminated in fact. This of course would make a huge difference in learning rates in the context of our model.

The absence of theta in the bat is a fascinating case. Although we cannot purport to know why this is, we find that it actually may be consistent with the theoretical mechanism for learning rates we put forth here. Namely, to attain a high learning rate the AC of place-cell activity should be modulated on a timescale commensurate with the window for plasticity. In the case of land mammals like rats and mice, the behavioral time-scale related to spatial navigation is too slow, therefore necessitating oscillatory modulation in the form of theta. In the case of flying animals, like the bat, changes in sensory input alone are already likely to occur on the order of tens to hundreds of ms due to the high velocity of flight, thereby obviating the need for theta. Note, for example, the two sample cells shown in Figure 4 of (Yartsev and Ulanovsky, 2013), which have AC which decay on the order of 100s of ms. We are not aware of data on STDP from bat hippocampus, so this is just an idea without any solid support. But in principle this idea could be put to the test.

Thank you for the reference.

We have now added several sentences to the Discussion section on the bat, essentially a slight rewriting of the above paragraph, as well as the above reference.

Reviewer #2:Summary:[…] The science behind the paper is very solid, and the analytical treatment in addition to the numerical simulations make this a very thorough, impressive modeling paper. The major criticisms revolve around how the authors chose to present some of their findings. Certain claims and findings are very exaggerated or misleading (see Major Critiques 1-3), and it is unclear what brain region are actually modeled (see Major Critique 4). Finally, there is an unsubstantiated claim about the model and oversight of extremely similar previously published work in the Discussion.Major Critiques:1) The authors consistently refer to the frequency band which optimizes the learning rate in the model as "theta", which is typically referred to as a hippocampal oscillation within the 8-12 Hz range. However, Figure 4A appears to show that optimal synaptic weight changes occur over a much broader frequency range, and Figure 4B specifically shows that optimal replay occurs within the 2-8 Hz range. The authors should be clear in the introduction about what is typically considered the theta range, and that, while their model does demonstrate the utility of oscillations within the theta range, the effect they observe is not limited to theta.

This is a fair point. The exact range which maximizes the learning rate clearly depends on the model parameters. As explained above in more detail in the section “Frequency dependence of the STDP rule”, the maximum learning rate occurs for *f* = 1*/*(2*π*√*τ*_+_*τ*−. The square-root dependence makes the optimal frequency only weakly sensitive to large changes in the STDP window, so that a large change in the width from 10s to 100s of ms only shift the optimal frequency from about 10Hz to 1Hz. Nearby frequencies do nearly as well as the optimal so that frequencies in the range 1-10Hz in general do very well.

In the Introduction we state that “for an intermediate range, set by the timescale of the plasticity rule, it [the growth rate] is higher…”. Also, when discussing Figure 4 we now replace “theta range” with “intermediate range”.

2) The authors should spend some time interpreting the analytical result from Equation 5 which demonstrates that "the growth rate of the even mode is found by multiplying the AC of place-cell activity by the window for plasticity, and integrating." No real physiological/functional interpretations/implications of this result are mentioned (which should always be done), and this leaves it unclear to readers (particularly ones without a rigorous computational background) how impactful this result actually is.

We of course want to provide a clear biological interpretation. This is given above in the section “Discussion of Equation 5”.

It is unsurprising that integrating the autocorrelation of place-cell activity multiplied by the plasticity window provides the expected growth rate. This is essentially saying that by considering all of the place-cell activity which occurs regularly, and the corresponding weight change given by STDP, while ignoring the noisy activity (which would tend to cancel out on average from this type of unsupervised learning rule) it is possible to estimate the growth rate of the even mode (i.e. the one relevant to learning and which governs replay).

Again, the detailed explanation is given in “Discussion of Equation 5” but we would disagree that this result is unsurprising or in any way obvious. What is clear, is that given a pair of neurons, the changes in the synaptic weights will depend on the cross-correlation (CC) of the neuronal activity. This follows just from the construction of the pairwise STDP rule. Adding up the potentiations and depressions from each spike pair separately (vis the STDP rule) is mathematically equivalent to taking the full spike trains, calculating the CC of the spikes and multiplying this by the STDP window. But here we are not talking about the weight changes between pairs of neurons. Rather we are talking about the growth of a pattern of synaptic connectivity which spans the entire network and which could involve changes in potentially hundreds or thousands of neurons simultaneously. The fact that one can infer the growth of a global pattern of connectivity just by looking at the AC of neuronal activity, a purely local quantity, is surely not trivial. It depends on the key assumption that the connectivity between any two place cells should only depend on the difference on their place cell positions, and not on their absolute positions. Again, details are given in “Discussion of Equation 5”.

The straightforward intuition of this result should be discussed both so that less computationally inclined readers can appreciate the result, and so that it does not mistakenly get taken to be predictive of a novel interpretation of how STDP leads to learning.

Again, we personally don’t find it straightforward, but we certainly agree that the explanation, both mathematical and physiological, should be as clear as possible, please see the section entitled “Discussion of Equation 5”.

3) The authors propose an unsubstantiated mechanism for how their network could also generate reverse replay in addition to the more commonly studied forward replay. They should either provide supporting data for this claim or go into a much more detailed explanation of their proposed mechanism.

As far as we know there are no *substantiated* mechanisms for how either forward or backward replay are generated. Everything is quite hypothetical at this point. But the point is well taken. We actually were conflicted about how much detail to go into on backward replay. The conundrum of course is that asymmetric connectivity, which is almost unavoidable in our model, leads to replay in one direction only. With symmetric connectivity there is no problem, and this can happen if the animal explores CW and CCW in equal measures. On a linear track this would not be possible due to global remapping, so the problem would persist. Our proposed solution is illustrated in simulations in Figure 2—figure supplement 7. A detailed description of the problem and our proposed solution is found above in section “Reverse replay”.

To begin with, it is unclear what exactly is meant by "synapses to downstream neurons become rapidly depressed". Is this referring to long-term depression, short-term presynaptic depression, etc.? Assuming short-term presynaptic depression (as it has the appropriate timescale) this mechanism would only make sense if the same place-cell sequences were activated during the theta cycle as those immediately reactivated in during sharp-wave replay. However, as far as I know, no such tight temporal correlation has been observed.

Indeed, it is just the short-term depression we have in the model (through the synaptic resource variable *x*). The depression is caused by purported local sensory input during awake quiescence, which would therefore be correlated with replay in that environment. Therefore our mechanism would not work during sleep for example. Details are given in the section “Reverse replay” above.

4) The authors need to be clearer from the beginning which region of the brain is modeled. The paper does not spend any time on this or the general architecture of the network model in the Introduction. The constant reference to "place-cells" implies that this is a model of CA1, however the connectivity of the model makes this more likely to reflect CA3; a fact which the authors bring up themselves in the Discussion. Their defence of how these results can still apply to learning and replaying place-cell sequences is solid, and I would recommend the authors take this approach from the start. Bringing it up at the end of the paper seems somewhat misleading and could potentially turn off readers and make them dismissive of the interesting findings reported in the paper.

We have now added several sentences to the Introduction and the beginning of the Results sections to make it clear we are discussing a model of a strongly recurrent network, such as in CA3.

Reviewer #3:This paper presents a theoretical study how STDP together with short-term synaptic depression and correctly tuned oscillation leads to replay like activity of hippocampal place cells. I think that none of the particular mechanisms are new, but the combination and its application to hippocampal sequences is and the analysis of the complete system is nice.- Were any special settings needed to make sure the rotation of the attractor stopped (Figure 2C bottom), and did not continue rotating? Or would sufficient experience lead to longer and longer sequences?

Excellent question. The duration of the burst does indeed depend on parameter values, in particular the mean external drive to the network. The dependence of the burst profile on the external input is shown in Figure 2—figure supplement 8. So there is only a small range of inputs for which the burst duration is physiological. Whenever the animal stops we set the input to this value. If we set it to another value for which the burst is longer lasting, there is no significant effect on the learning process; it’s just an aesthetic choice which gives rise to sharp-wave-like profiles.

- The maximum in Figure 4A hinges on the lack of frequency dependence in the STDP rule. The problem is that, AFAIK, STDP always has shown frequency dependence (more LTP at higher frequencies) and it is hard to imagine how biophysically a precise balanced STDP model could be constructed. Currently it is written as if this is a minor issue, and that STDP needs some stabilization anyway. That might be true but the maximum might be lost. I think this weakness of the model needs to be exposed more.

Of course we don’t expect an in-vivo STDP rule to be precisely balanced, some additional plasticity mechanisms are surely present with homeostatic effect. On the other hand, the frequency-dependence observed in-vitro would be disastrously unstable in an in-vivo network, an issue noted by (Zenke et al., 2015) who suggested heterosynaptic and homeostatic mechanisms to counter this. However, we completely agree that with a sufficiently strong frequency dependence (we call it rate dependence so as not to confuse it with the modulation frequency *f*) the pairwise rule becomes essentially irrelevant and the peak would be lost. A detailed discussion of this can be found above in the section “Frequency dependence of the STDP rule”, including changes made to the text.

Secondly, related but a bit less important, I would expect that even with balanced STDP the weights diverge due to the correlation contribution to LTP.

In our model this would not happen. But if the reviewer means in a spiking network in which pairwise spike correlations (above and beyond those due to underlying rate variations) are not negligible, this may be true. If correlations are weak then the divergence would be slow, and could be brought into check by a slow homeostatic process. Furthermore as noted in (Graupner et al., 2017), realistic in-vivo-like firing patterns “imply low sensitivity of synaptic plasticity to spike-timing as compared to firing rate”. This suggests that in a recurrent spiking network in the asynchronous state, learning due to variations in firing rates, which is the mechanism we study here, should be much faster than changes due to spike correlations.